# The impact of inhomogeneous emissions and topography on ozone photochemistry in the vicinity of the Hong Kong island

Yuting Wang[1,*], Yong-Feng Ma[2,*], Domingo Muñoz-Esparza[3], Cathy W. Y. Li[4], Mary Barth[3], Tao Wang[1], Guy P. Brasseur[1,3,4]

[1]Department of Civil and Environmental Engineering, the Hong Kong Polytechnic University, Hung Hom, Kowloon, Hong Kong
[2]Department of Mechanics & Aerospace Engineering, Southern University of Science and Technology, Shenzhen, 518055, China
[3]National Center for Atmospheric Research, Boulder, CO, USA
[4]Max Planck Institute for Meteorology, 20146, Hamburg, Germany
[*]These authors contributed equally to this work.

*Correspondence to*: Yuting Wang (yuting.wang@polyu.edu.hk)

**Abstract.** Global and regional chemical transport models of the atmosphere are based on the assumption that chemical species are completely mixed within each model grid box. However, in reality, these species are often segregated due to localized sources and the influence of the topography. In order to investigate the degree to which the rates of chemical reactions between two reactive species are reduced due to the possible segregation of species within the convective boundary layer, we perform large-eddy simulations (LES) in the mountainous region of the Hong Kong island. We adopt a simple chemical scheme with 15 primary and secondary chemical species including ozone and its precursors. We calculate the segregation intensity due to inhomogeneity in the surface emissions of primary pollutants and due to turbulent motions related to topography. We show that the inhomogeneity in the emissions increases the segregation intensity by a factor 2-5 relative to a case in which the emissions are assumed to be uniformly distributed. Topography has an important effect on the segregation locally, but this influence is relatively limited when considering the spatial domain as a whole. In the particular setting of our model, segregation reduces the ozone formation by 8-12 % compared to the case with complete mixing, implying that the coarse resolution models may overestimate the surface ozone when ignoring the segregation effect.

## 1 Introduction

The spatial distribution of reactive species in the atmosphere derived by global or regional chemical-meteorological models is obtained by solving a system of nonlinear continuity (mass conservation) equations coupled with the Navier-Stokes (momentum) and energy conservation equations (Brasseur and Jacob, 2017). In most cases, a numerical approximation of the solution of these partial differential equations is found at a finite number of locations on a grid that covers the three-dimensional

(3D) geographical domain under consideration. The size of the spatial patterns that is explicitly resolved by such models is determined by the size of the adopted grid meshes. Smaller features, called subgrid-scale (SGS) processes that influence the large-scale dynamical and chemical solutions, are often represented by closure relations based on empirical parameterizations.

With the computer resources currently available, the spatial resolution adopted for the discretization of the model equations is typically of the order of 50-100 km in the case of global models and 1-50 km in the case of regional models used for operational numerical weather and climate predictions. Thus, in both cases, small-scale processes such as turbulent motions in the boundary layer, mountain flows, sea breeze, urban dynamics, shallow clouds, as well as the complexity of surface chemical

emissions are crudely represented or parameterized. Coarse models, for example, assume total mixing between trace species inside each grid mesh, and therefore do not accurately account for the segregation that may exist between these species in turbulent flows. The segregation effect is important for fast reactions, of which the chemical timescale is shorter than the turbulent timescale. In this case the reactants remain segregated rather than reacting, and such segregation tends to reduce the averaged rate at which chemical reactions happen within a model grid mesh (Komori et al., 1991; Schumann, 1989). Previous

studies, such as Vilà-Guerau de Arellano and Duynkerke (1993) and Kramm and Meixner (2000), considered the segregation effect in the boundary layer parameterization in the chemical transport models and showed that it is important to take the segregation into account in current photochemical models.

Numerical treatment of the turbulent flow can be provided by Large Eddy Simulation (LES) models. LES is a rapidly evolving

approach for modelling the turbulent flows, which was initially proposed by Smagorinsky (1963) and first explored by Deardorff (1970). In this approach, the unsteady Navier-Stokes and continuity equations are filtered to remove the smallest eddies, while capturing the eddy motions at a size larger than a specified cut-off width. The interactions of the larger, resolved eddies and the smaller, unresolved eddies are addressed by specifying a SGS stress model (Deardorff, 1970; Smagorinsky, 1963).


The LES technique has been used in past studies to quantify the segregation effect in the turbulent atmospheric boundary layer. Schumann (1989) simulated a single reaction in the convective boundary layer with one bottom-up and one top-down tracer, and showed that the segregation of one reaction is dependent on the ratio of the chemical and turbulent timescales, the concentration ratio of the two reactants, and the initial condition. Patton et al. (2001) used LES to simulate scalars that are

emitted from forest canopy with different decay rates, and presented the influence of chemical reactivity on the scalar distribution, variance, vertical flux, which affect the segregation. Other LES studies with different configurations of surface emissions pointed out that spatially inhomogeneous emissions influence considerably the segregation intensities. These studies, such as Krol et al. (2000), Auger and Legras (2007), and Ouwersloot et al. (2011), mainly focused on the isoprene chemistry, and compared the segregation intensities of the reaction between isoprene and hydroxyl radical (OH) with

homogenous and inhomogeneous isoprene emissions, and showed that the heterogeneity in the emissions largely increases the

segregation intensities. There are other factors that affect the segregation intensities. For example, Li et al. (2016) investigated the sensitivity of segregation of volatile organic compounds (VOC) to weather conditions (e.g., temperature, humidity and the presence of clouds). They showed that isoprene segregation is largest under warm and convective conditions. Kim et al. (2016) showed that the segregation intensity of isoprene and OH differs at low and high nitrogen oxide ($NO_X$) levels caused by the primary production and loss reactions of OH under different $NO_X$ regime. Li et al. (2017) added the effect of aqueous-phase chemistry in a LES model and showed that segregation of OH and isoprene is enhanced in clouds. Studies focusing on isoprene chemistry in a forested region have been initiated to understand the underestimation of the OH in global models (Dlugi et al., 2019; Ouwersloot et al., 2011). These studies all used flat domains and calculated the domain-averaged segregation intensities to account for the errors induced by the turbulence in regional or global models. However, the impact of the terrain on the segregation was not considered. Previous studies showed that complex terrain has an important impact on the turbulence structure in the boundary layer (e.g., Cao et al., 2012; Rotach et al., 2015; Liang et al., 2020) and on the evolution with time of the boundary layer height (De Wekker and Kossmann 2015). Therefore, the segregation intensity is expected to be affected by the topography.

This paper uses a LES model included in the Weather Research and Forecasting (WRF) model (Skamarock et al., 2008; 2019) to investigate the importance of segregation between reacting species in an area where surface emissions are spatially very inhomogeneous and the flow is turbulent under the influence of a complex topography. Under such conditions, the vertical mixing and the reaction rates between chemical species in the boundary layer are expected to be sensitive to the strength of the large eddies. The simulations are performed in a geographical area covering the island of Hong Kong. The surface wind measurements from the Hong Kong Observatory (HKO) station show the prevailing wind blow from the east in about 80 % of the time and from the west about 15 % of the time (Shu et al., 2015). There are two important features in the air pollution in Hong Kong: the pollution sources are concentrated in the very dense urban region, mostly along the coast, while natural emissions occur in large forested areas in the center of the island; both regions are separated by the complex topography. With the intense and inhomogeneous emissions in such urban environment, the resultant segregation can cause a large impact on the calculation of chemical reactions (Li et al., 2020). This impact is expected to be even larger with the influence of complex topography. In this study, we therefore set up a domain with a mountainous terrain characterized by complex flows determined by the topography and the occurrence of related turbulent motions.

The purpose of the study is to investigate the importance of interactions between chemistry and turbulence in the planetary boundary layer (PBL), as well as to assess how the spatial segregation between chemical species affects the nonlinear production and destruction rates of key chemical species including ozone. Since the study is intended to be conceptual, we adopt a simple chemical scheme to represent the chemical interactions between ozone and its precursors. We make plausible assumptions about the spatial distribution of the surface emissions of primary species emitted in the forested and urbanized areas of the island. We estimate how the vertical eddy transport fluxes of the chemical species and the covariance between

their concentrations, generally unaccounted for in coarse atmospheric models, affect the distribution of chemical species in the lowest levels of the atmosphere in the vicinity of the island.

The present modelling study should be viewed as a step towards a more complex and realistic investigation of chemical and turbulent processes occurring in the densely populated urban area of Hong Kong characterized by a complex urban canopy of high-rise buildings and street canyons built on an uneven topography surrounded by the ocean and affected by emissions in Mainland China and other Asian countries and by the presence of an active harbour with intense shipping in the region. The present study focuses on the impact of the heterogenous emissions and the turbulent flow generated by the topography on chemical processes. Model description and dynamical and chemical settings are introduced in Section 2. Results of the large-eddy simulations and the interpretations of the model output are presented in Section 3. Section 4 provides the principal conclusions of this study.

## 2 Methodology

### 2.1 Model description

The WRF model (version 4.0.2) with the ARW (Advanced Research WRF) core (Skamarock et al., 2008; 2019) was used to perform mesoscale meteorology simulations that provided the initial fields for the large eddy simulations. The LES module included in WRF was run in an idealized mode as implemented and evaluated by Moeng et al. (2007), Kirkil et al. (2012), and Yamaguchi and Feingold (2012). A low-pass filter was applied to separate the large and small eddies, where the large eddies (energy-injection scales near the classic inertial range of 3D turbulence) were explicitly resolved, while the small eddies were parameterized by the SGS model. In the idealized LES, the initial physical conditions were specified by uniform values over the entire domain; a random perturbation was imposed initially on the mean temperature field at the lowest four grid levels to initiate the turbulent motions.

### 2.2 Dynamical settings

A nested two-domain setup was adopted for the LES simulation. The size of the outer domain was 45 km × 45 km and the spatial resolution was 300 m.  The size of the inner domain was 24 km × 24 km with a horizontal grid spacing of 100 m. The vertical layers were the same for both domains with 100 vertical levels, and the model top was set at 4 km altitude. Double periodic boundary conditions were used in both the west-east and south-north directions for the outer domain. One-way nesting was used in which the outer domain provided turbulence-inclusive boundary conditions for the inner domain (Moeng et al., 2007; Muñoz-Esparza et al., 2014).

The initial profiles for the potential temperature ($\theta$) and water mixing ratio ($q$) for the outer domain were taken from the output of the mesoscale WRF run operated at a spatial resolution of 1.33 km (shown by the green dash line in Figure 1), and were

interpolated to the 40 m vertical grid spacing with the highest level located at 4 km. The date (Aug 1, 2018) and time of the day (UTC 04:00; local time 12:00) were chosen to correspond to a typical summer condition in Hong Kong with a well-developed convective boundary layer. Four cases were considered with initial winds blowing uniformly from the west (TERW; TER stands for terrain), the east (TERE), the south (TERS), and the north (TERN), respectively. The initial wind speed in each case was equal to 10 m s$^{-1}$.

The LES version adopted here used the Deardorff's turbulent kinetic energy (TKE) scheme to compute the SGS eddy viscosity and eddy diffusivity for turbulent mixing. The Coriolis parameter was set according to the latitude of the Hong Kong island. The Kessler microphysics scheme (Kessler, 1969) was used to derive cloudiness in the model. The WRF's radiation, land surface, and PBL schemes were all turned off. We applied a fixed sensible heat flux of 230 W m$^{-2}$ at the bottom boundary, while the latent heat flux was calculated in the model from the surface water vapor content specified at the beginning of the simulations. We chose to apply a relatively small sensible heat flux (compared to summer noon conditions in Hong Kong) in order to produce a gradual development of the boundary layer and keep the convection condition unchanged during the course of the simulation. With this choice, the buoyancy flux was dominated in the simulation by the kinematic sensible heat flux, which is about 2~3 times larger than the kinematic moisture heat flux.

The outer model domain was assumed to be entirely flat, while the inner domain included a representation of the topography of the Hong Kong island. This modelling setup followed the approach of Kosović et al (2014). The elevation of the surface was taken from the ALOS world 3D data (Takaku et al., 2014) distributed by OpenTopography (https://opentopography.org; last access: Jun 1, 2020) with a spatial resolution of 30 m. To avoid numerical errors due to the terrain following coordinate, the terrain was somewhat smoothed in areas where the surface slope exceeds 25 º. The smoothed terrain height is shown in Figure 2a. To simplify the simulation and to focus on the terrain shape instead of the terrain types, the surface of entire domain was set as land to ignore the influence of the sea.

### 2.3 Chemical settings

A simple $O_3$-$NO_X$-VOCs chemical mechanism with 15 reactive species and 18 photochemical reactions (see Table 1) was adopted in this conceptual study and is based on the simple scheme for ozone production from hydrocarbon oxidation described in the textbook by Brasseur and Jacob (2017). We included two primary hydrocarbons, RH-A and RH-B, which represent anthropogenic (labelled -A) and biogenic (labelled -B) VOCs, respectively. RH-A was treated as a surrogate for propane and RH-B as a surrogate for isoprene. The corresponding rate constants ($k$) for the oxidation by the hydroxyl radical OH at temperature of 300 K are equal to $1.1 \times 10^{-12}$ cm$^3$ molecule$^{-1}$ s$^{-1}$ and $1.0 \times 10^{-10}$ cm$^3$ molecule$^{-1}$ s$^{-1}$, respectively. We can derive the lifetime of RH-A and RH-B by the expression $1/(k \times [OH])$ with the respective values of the rate constants $k$. With an OH concentration ([OH]) of $5 \times 10^6$ molecule cm$^{-3}$, the corresponding chemical lifetimes of RH-A and RH-B are approximately 2 days and 30 minutes, respectively.

Among all the species, NO, CO, RH-A, and RH-B were emitted at the surface. The background conditions of the polluted atmosphere were represented by assuming emission rates in the outer domain to be uniform with values of $1.2 \times 10^{12}$ molecule cm$^{-2}$ s$^{-1}$, $8.0 \times 10^{12}$ molecule cm$^{-2}$ s$^{-1}$, and $7.0 \times 10^{11}$ molecule cm$^{-2}$ s$^{-1}$, for NO, CO, and RH-A respectively. The RH-B emission was set to be $3.0 \times 10^{11}$ molecule cm$^{-2}$ s$^{-1}$ for the condition in Hong Kong with large tropical forested area. For the inner domain, we considered two specific regions corresponding to urban and forested areas, based on information provided by the land use

map from ESA CCI (ESA, 2017, https://www.esa-landcover-cci.org; last access: Jun 1, 2020). The resulting emission map on the nested domain that includes terrain features is shown in Figure 2b with NO, CO, and RH-A emitted in the urban region and RH-B emitted in the forested region. The emission rates in the inner domain were calculated by dividing the corresponding values adopted in the outer domain by the area fraction of the corresponding land use type, so that the averaged emission rates for the whole inner domain were the same to those of the outer domain. In order to separate the influence of the emissions and

topography, two additional simulations were conducted. A simulation with homogeneous emissions and without topography for both domains (HOMF; HOM stands for homogeneous emission; F stands for flat terrain) was performed as a baseline experiment. To assess the role of the inhomogeneous emissions on our results, a simulation with flat terrain but with inhomogeneous emissions as shown in Figure 2b (referred to as HETF; HET stands for heterogeneous emission; F stands for Flat terrain) was also conducted. The details of all the experiments are listed in Table 2.


An atmospheric destruction of reservoir species (HNO$_3$, H$_2$O$_2$, ROOH-A, ROOH-B) was applied to balance the surface emissions of the primary species. The removal of these reservoirs by photolytic processes in the atmosphere is slow (more than 2 weeks in the case of HNO$_3$ and several days for peroxides) so that most of the loss is due to wet removal or dry deposition. The model did not consider the possible occurrence of convective precipitation as often observed during

summertime in Hong Kong, and hence no detailed formulation was used for the wet removal in the LES model. However, in order to keep a balance (stationary state) in the background concentrations and avoid an accumulation of species produced from the ongoing emissions, the removal of the soluble species HNO$_3$, H$_2$O$_2$, ROOH-A, and ROOH-B was occurring with a first order rate of $2.5 \times 10^{-5}$ s$^{-1}$ (lifetime of about half a day). This lifetime may be viewed as representing the mean time period separating successive convective rain events during summertime. For the dry deposition on the surface, we adopted for the

grass/forested areas outside the urbanized regions values based on the measurements of Wu et al. (2011) and on the analysis of Ganzeveld and Lelieveld (1995). The values were reduced over urban areas. The detailed deposition velocities for the different species and the different land types are provided in Table 3.

In order to generate reasonable initial profiles for the chemical species for the two-domain simulations and to reduce the spin

up time, a simple one-domain LES simulation was run for two days using the same chemical scheme as in the two-domain simulation. The one-domain LES used the same homogeneous emissions as described above. The time-varying photolysis rates were calculated by the TUV scheme for producing the diurnal variation in the photochemistry. The domain-averaged

profiles for the chemical species at the local time 12:00 of the second day were then used as the chemical initial profiles for the two-domain simulations. The initial profiles are shown by green lines in Figure 3.

## 200    2.4 Formulations of chemistry-turbulence interactions

Following the concept of the Reynolds decomposition, any physical variable $A$ such as the concentration of chemical species, can be expressed as the sum between its mean value $\langle A \rangle$ (here $\langle\ \rangle$ stands for time average) and the fluctuation $A'$ caused, for example, by turbulent motions:

$$A = \langle A \rangle + A' \tag{1}$$

For a second-order chemical reaction

$$A + B \rightarrow C \tag{2}$$


with the reaction rate constant,

$$k = k_0 e^{-\frac{T_0}{T}} \tag{3}$$

where $k_0$ and $T_0$ are reaction-dependent constants, the rate $R$ of the reaction is expressed as

$$R = k \cdot A \cdot B \tag{4}$$

If we ignore the temperature fluctuation (i.e., $T' = 0$), the averaged reaction rate is

$$\bar{R} = k \cdot \langle A \cdot B \rangle = k \cdot \langle A \rangle \cdot \langle B \rangle + k \cdot \langle A' \cdot B' \rangle = k \cdot \langle A \rangle \cdot \langle B \rangle \cdot (1 + I_{AB}) \tag{5}$$

where $\langle A' \cdot B' \rangle$ is the chemical covariance between the two reacting chemicals and $I_{AB}$ is called the segregation intensity (Danckwerts, 1952) defined as

$$I_{AB} = \frac{\langle A' \cdot B' \rangle}{\langle A \rangle \cdot \langle B \rangle} \tag{6}$$

The intensity of the segregation is therefore equal to the covariance of the two reactants divided by the product of their mean concentrations. $I_{AB}$ is equal to zero when the chemicals $A$ and $B$ are fully mixed, and equals to -100 % when the two chemicals

are fully segregated. For initially segregated $A$ and $B$, the corresponding segregation intensity is closer to -100 % if the chemical

reaction between the two species is fast. $I_{AB}$ is positive when the concentrations of the two chemicals are correlated.

The intensity of segregation between two species $A$ and $B$ is related to the Damköhler number ($Da$) (Damköhler, 1940):

$$Da_{A(B)} = \frac{\tau_{turb}}{\tau_{chem,A(B)}} \tag{7}$$


where $\tau_{turb}$ is the turbulent timescale (typically 10 minutes during daytime; considerably longer during night time), and $\tau_{chem,A(B)}$ is the reaction timescale of $A$ when reacting with $B$. If the chemical timescale of $A$ is shorter than the turbulent timescale ($Da$)1; called *fast chemistry limit*), the reaction between $A$ and $B$ is limited by the rate at which turbulent motions bring reacting species together; the covariance between fluctuating components as well as the segregation intensity are high.

This occurs when the reaction rate constant $k$ is large (Schumann, 1989; Vinuesa and Vilà-Guerau de Arellano, 2005; 2011) or when the emission of $A$ or $B$ is intense (Molemaker and Vilà-Guerau de Arellano, 1998; Kim et al., 2016; Li et al., 2020). Other factors such as inhomogeneous emissions (Ouwersloot et al. 2011; Auger and Legras, 2007; Li et al., 2020) and other structures that obscure mixing also result in a more negative segregation intensity. Under this situation, coarse models such as regional chemical transport models may not provide accurate results since they ignore the influence of sub-grid scale turbulence

on rate at which a reaction between $A$ and $B$ occurs. When the Damköhler number $Da \ll 1$ (called *slow chemistry limit*), the chemical species $A$ under consideration is well mixed with small covariances with reactant $B$ and with no significant segregation. In this case, the reaction is controlled by chemistry and its rate is proportional to the product of the mean concentrations. This represents a case in which the rate of chemical reactions is well represented in coarse models. When the Damköhler number is close to 1, the interaction between chemistry and turbulence is strong as the chemical species are equally

controlled by chemistry and turbulence.

In the Reynolds decomposition described here, the reaction rate is thus expressed by the sum of two terms: the first term is proportional to the product of the mean concentrations, represented, for example, by the concentration averaged over a grid cell in a global and regional model. The second term accounts for the contribution of subgrid chemical-turbulent interactions,

and can be estimated using a large-eddy simulation. The effective rate constant $k_{eff}$ of a reaction affected by turbulent motions is therefore (Vinuesa and Vilà-Guerau De Arellano, 2005)

$$k_{eff} = k(1 + I_{AB}) \tag{8}$$

Thus, a negative value of the segregation intensity $I_{AB}$ tends to reduce the average rate at which a reaction occurs, while a positive value of $I_{AB}$ leads to an enhancement in this rate.

The segregation intensity can also be written as a function of the correlation coefficient and the concentration fluctuation intensity as shown in the paper by Ouwersloot et al. (2011). The standard deviation of $A$ is expressed as $\sigma_A = \langle A' \cdot A' \rangle^{\frac{1}{2}}$ and the covariance of $A$ and $B$ is $\sigma_{AB} = \langle A' \cdot B' \rangle$, so the segregation intensity becomes

$$I_{AB} = \frac{\sigma_{AB}}{\sigma_A \cdot \sigma_B} \cdot \frac{\sigma_A}{\langle A \rangle} \cdot \frac{\sigma_B}{\langle B \rangle} = r \cdot i_A \cdot i_B \qquad (9)$$

where

$$r = \frac{\sigma_{AB}}{\sigma_A \cdot \sigma_B} \qquad (10)$$

is the correlation coefficient of $A$ and $B$, which determines the sign of the segregation intensity, while

$$i_A = \frac{\sigma_A}{\langle A \rangle} \qquad and \qquad i_B = \frac{\sigma_B}{\langle B \rangle} \qquad (11)$$

defined as the concentration fluctuation intensity of $A$ and $B$, controls the strength of the segregation. In this paper, the mean fields of the chemical species were all calculated as one-hour time average at given points in the domain.

## 3 Results

The simulations for the inner domain were analysed and are shown in this Section. The results of the simulations are presented in several subsections. First, we derive domain averaged characteristics of physical and chemical quantities (i.e., temperature, humidity, concentrations of chemical species) from a baseline experiment (HOMF) that ran with uniform surface emissions in the absence of topography. Second, we analyse the effects of inhomogeneous (spatially concentrated) surface emissions (HETF) on the distribution of chemical species and on the spatial segregation between reacting species. Third, we discuss the impact of the topography on the same quantities. Finally, we assess the influence of different mean wind directions on our model results.

### 3.1 General development

The time evolution of the profiles of the potential temperature and water vapor mixing ratio (averaged over the entire domain) is shown in Figure 1. The dash lines represent the output from the mesoscale WRF model at a spatial resolution of 1.33 km. The green dash lines at UTC 04:00 (local time 12:00) represent the initial profiles adopted for the LES simulation, and the magenta dash lines are the WRF output at UTC 05:00 (local time 13:00). At this time, the temperature has increased in the

boundary layer and the top of the PBL has been lifted in response to the warming of the surface. The sounding measurements at UTC 05:00 at King's Park station are used to validate the simulated physical quantities (black lines in Figure 1). The mesoscale WRF model reproduces the potential temperature quite well, while it underestimates the water vapor mixing ratio, especially in the boundary layer. This may be related to a simulated weaker southerly wind, which results in relatively less water vapour transport from the South China Sea. The red and blue solid lines show the large eddy simulation (HOMF) after two and four integration hours, respectively. The simulated water vapor from LES is higher than in the mesoscale model, and the agreement with the measurements is improved, especially at high altitudes. The convective boundary layer height is detected by the virtual potential temperature ($\theta_v$) gradient method, which is defined as the height where $\frac{\partial \theta_v}{\partial z} = \frac{\theta_v(z) - \theta_{v_s}}{z - z_s}$ (the subscript "$s$" represents the surface) first exceeds a threshold value (Liu and Liang, 2010). Due to the constant surface heat flux forcing, the PBL height gradually increases with a trend of ~0.028 m s$^{-1}$. The PBL height is about 797 m and 985 m at hour 2 and hour 4, respectively. This deepening tendency of the PBL is the same as in the mesoscale WRF simulation, but the growth in the PBL height in LES is slower than that in WRF because of the small surface heat flux adopted in the LES model. It should be noted that the derived PBL heights are sensitive to the estimation methods (e.g., Seidel et al., 2010; Li et al., 2019). The virtual potential temperature gradient method is chosen because the calculated PBL height is consistent with that derived in the mesoscale WRF. However, to be considered as realistic, the calculated PBL heights should be compared to a sufficient number of observations rather than mesoscale WRF estimates. In fact, the PBL processes and cloud convection are difficult to be represented in the mesoscale models (Mape et al., 2004; Barth et al., 2007), and the derived PBL heights are strongly affected by the cloud physics and PBL schemes adopted in the simulations (e.g., Li et al., 2019). The estimated turbulent timescale for the LES simulation is about 9 min, which is similar to timescales reported in previous studies (e.g., Anfossi et al., 2006).

Figure 3 shows the domain-mean profiles of several chemical species at hours 2 and 4. The green lines represent the initial profiles applied in the LES model. The profiles at hours 2 and 4 are very similar, which suggests that stationary conditions have been reached in the LES simulation. The atmospheric concentrations of the species emitted at the surface (NO, RH-A, RH-B) are largest near the ground, and decrease with altitude. Because $O_3$ is consumed by NO and since NO is highest at the surface, the $O_3$ concentration is lowest in the bottom layers of the model. The same situation exists for OH that reacts with the primary species CO, RH-A, and RH-B that are emitted at the surface. $NO_2$ is produced by the reaction between NO and $O_3$; this reaction is therefore largely dependent on the NO concentration, so that the $NO_2$ shows a similar profile to that of NO. $RO_2$-A and $RO_2$-B are produced by the oxidation by OH of RH-A and RH-B; however, RH-A reacts more slowly than RH-B, and therefore $RO_2$-A vertical profile is more affected by the concentration of OH than $RO_2$-B that shares the same structure as that of RH-B.

In order to assess the average effect of turbulence on the reaction rates, we calculated the segregation intensities ($I_{AB}$) using
Equation (6) and averaged them over the entire domain (shown in Figure 4) for the following 5 reactions:

$$\text{R8:} \quad NO + O_3 \rightarrow NO_2 + O_2$$
$$\text{R16:} \quad RH\text{-}A + OH \rightarrow RO_2\text{-}A$$
$$\text{R17:} \quad RO_2\text{-}A + NO \rightarrow HCHO + HO_2 + NO_2$$
$$\text{R19:} \quad RH\text{-}B + OH \rightarrow RO_2\text{-}B$$
$$\text{R20:} \quad RO_2\text{-}B + NO \rightarrow HCHO + HO_2 + NO_2$$

Among these reactions, R8 plays a key role in linking the atmospheric levels of tropospheric ozone and nitrogen oxides; R16 and R17 represent the degradation path of anthropogenic VOCs while R19 and R20 account for the same reactions, but for
biogenic VOCs. In this study, we only analyse the segregation effect in the boundary layer up to 800 m to avoid the more complex influence from clouds that are formed higher up in the atmosphere. Using equation (7), we derive in the daytime boundary layer Damköhler numbers $Da(NO)$ and $Da(O_3)$ of 9.8 and 0.5 respectively for reaction R8. The Damköhler numbers for the degradation of primary anthropogenic and biogenic hydrocarbons RH-A and RH-B by the hydroxyl radical OH are 0.004 and 0.3, respectively. This indicates that the oxidation of the anthropogenic hydrocarbon (surrogate of propane) is slow,
while it is considerably faster in the case of the biogenic hydrocarbon (surrogate of isoprene). The value of 0.3 calculated for RH-B is in the range of 0.01 – 1.0 derived by previous studies of isoprene (e.g., Patton et al., 2001; Vinuesa and Vilà-Guerau de Arellano, 2005; Li et al., 2016; Dlugi et al., 2019). The Damköhler numbers for the reaction of peroxy radicals with nitric oxide, $Da(RO_2\text{-}A)$ and $Da(RO_2\text{-}B)$, are equal to 188 and 248 respectively, which indicates that the reaction of the organic peroxy radicals with NO is fast for both species from anthropogenic and biogenic origins (Verver et al., 2000). The calculation
of the Damköhler number is sensitive to several factors including the weather conditions and the concentration of the species (Li et al., 2016). As a result, the calculated values of $Da$ can vary over a wide range and change as one adopts a different chemical mechanism. The horizontal averaged segregation profiles for the selected reactions from surface to 1000 m are shown in Figure 4. For the LES experiment with flat terrain and homogenous emissions, the segregation is weak near the surface, and becomes larger at higher altitudes. It is generated by the turbulent patterns of the flow. The segregation intensities for reactions
R8, R16, R17, and R19 are negative, which highlights the anti-correlation between the atmospheric concentration of the reactants. The segregation intensity, however, is positive in the case of R20 because $RO_2$ and NO are positively correlated. We calculated the statistics from the segregation fields for the center region of the domain ($14 \times 14$ km$^2$) so that we exclude the influence of the buffering zone near the lateral boundaries of the domain. We provide values for two altitude layers: 0-500 m and 500-800 m, as shown in Table 4. The separation at 500 m is adopted to make comparisons with subsequent simulations in
which the effect of the terrain is considered. The mean segregation intensity for the reaction between NO and O$_3$ is -0.60 % below 500 m and -0.95 % above 500 m, which are the smallest values among the five selected reactions. This results from the fact that the reaction rate between NO and O$_3$ is relatively small, as well as the rapid cycling between NO and NO$_2$. The

intensities for the reaction between the anthropogenic hydrocarbon (RH-A) and OH are -0.84 % at the lowest levels, and -1.52 % at the highest level, respectively. The reaction of the biogenic hydrocarbon (RH-B) is considerably faster than that of RH-A; as a result, segregation is as large as -5.08 % and -8.38 % for the low and high levels, respectively. The calculated segregation intensity for RH-B and OH is comparable to the values from previous studies, e.g., -7 % (PBL averaged) as reported for the homogeneous case by Ouwersloot et al. (2011) and -5 % to -6 % as found by Kim et al. (2016). Kim et al. (2016) showed that the segregation intensity between isoprene and OH vary with $NO_X$ levels. This results from the fact that the primary OH production and loss rates vary according to the $NO_X$ regime under consideration. Therefore, the differences between values reported by different studies can be attributed, at least in part, to the different $NO_X$ levels. Ouwersloot et al. (2011) used a low $NO_X$ condition, while our study considers a polluted situation with high $NO_X$ concentrations. In order to further compare with the previous studies, the relationship between segregation intensity and correlation coefficient ($r$) for the reaction between RH-B and OH is plotted in Figure 5. The calculated correlation coefficients are mostly in the range of -0.6 to -1.0, which is consistent with the model results from Ouwersloot et al. (2011); while they are larger than the measurements from some campaigns resulting from the measurement noise (Dlugi et al., 2019). The mean values calculated for NO and $RO_2$-A are -3.14 % and -5.38 %. For the reaction of the positively correlated species, NO and $RO_2$-B, the segregation intensities are +2.77 % and +5.43 %.

Figure 4 also shows that the negative segregation intensities are larger for hour 2 than for hour 4 of the simulation, while the positive segregation is smaller at hour 2 compared to hour 4. This highlights the enhanced mixing of the tracers as time proceeds, so that the reactions become increasingly effective.

## 3.2 Impact of inhomogeneous surface emissions on the chemical reactions

To investigate the impact of the spatially inhomogeneous emissions on Hong Kong island, a control run (HETF) was conducted using the emission distribution shown in Figure 2. The mean concentrations of several chemical species near the surface and in a vertical cross section along the west-east direction at latitude 22.275 ºN are shown in Figure 6. NO with anthropogenic emissions has the highest concentrations in the urban area at the edge of the Hong Kong island. RH-A shares the same pattern as NO, so that it is not shown here. Since RH-B has a biogenic source located in the forested region of the island, the highest values are found in the centre of the island. $NO_2$ produced by the reaction between NO and $O_3$ shows the same pattern as NO, while $O_3$, which is negatively correlated with NO, exhibits the lowest concentration in the urbanized area. OH, which is depleted by both anthropogenic and biogenic species, is characterized by low concentrations above the whole island. Since the anthropogenic emissions including CO and RH-A are considerably larger than the emissions of RH-B, the OH concentrations are lowest in the urbanized area. The OH concentration also shows peak values at the edge between the areas dominated by anthropogenic and biogenic emissions, where the destruction of the radical is smallest. The organic peroxy radicals of anthropogenic and biogenic origin, $RO_2$-A and $RO_2$-B, share the patterns found for OH and RH-B, respectively. This is consistent with the discussion in Section 3.1.

The segregation intensity derived for the HETF case is shown in Figure 7. Since the segregation intensities of the reactions of OH with RH-A and NO with RO$_2$-A have similar distribution to that of the NO and O$_3$ reaction, they are not shown in the figure. Different from the experiment HOMF with the homogenous emissions, the segregation intensity from the HETF run is largely dependent on the distribution of the emissions. The segregation map near the surface shows that intensity is much larger near the source region than at other locations. For the reaction of NO and O$_3$, the concentration distributions of the two species are opposite to each other, and the segregation effect is negative. Because of the stronger depletion of OH by the anthropogenic species (both CO and RH-A) than by biogenic RH-B, the OH concentration is lower in the urban region and higher in the forest area, which results in a opposite pattern to RH-A. Thus, the segregation effect is negative for the reaction between RH-A and OH. This is also the case for the reaction between NO and RO$_2$-A, since RO$_2$-A follows the pattern of OH. The segregation map for RH-B and OH is complicated with both positive and negative values, depending on the location; the positive segregation intensities are located mainly at the edge between urban and forested areas where OH concentration exhibits some peaks in its concentration. Because RH-B depletes OH, the two species are expected to be negatively correlated; however, OH is also consumed by other species, which affects the distributions of the radical. This implies that the heterogenous emissions not only lead to different intensity distributions, but they also can change the sign of the segregation when one chemical species reacts with several species present in different concentrations at separated locations. The correlation coefficient for the RH-B and OH is shown in Figure 5. Different from the simulation with homogenous emissions, the correlation coefficients vary between -1 and +0.7, which decide the sign of the segregation intensities. From the segregation map near the surface, we can see that the segregation intensities for NO and RO$_2$-B are negative, which is different from the HOMF case in which the emissions are homogeneous. In the HOMF simulation, the emissions of NO and RH-B (which is highly correlated to RO$_2$-B) are co-located, so NO and RO$_2$-B are positively correlated; while in the HETF simulation, the emission of RH-B is separated from the NO emissions, resulting in a negative correlation between NO and RO$_2$-B. The vertical patterns of the segregation intensity, shown in the second row of Figure 7, indicate that the impact of the emission distribution is substantial from the surface to about 500 m height, and is affected by the strength of convection. It is seen that the segregation for NO and RO$_2$-B is only negative at low altitudes; it becomes positive at higher levels where the impact of the separated sources has vanished. This is consistent with what is reported by Ouwersloot et al. (2011) and Li et al. (2020) for their cases with heterogeneous emissions. The large magnitude of segregation intensity between RO$_2$-B and NO near the surface is also comparable to the values reported in Li et al. (2020) in their cases with heterogeneous emissions.

The calculated mean segregation intensities for NO and O$_3$ are -3.21 % and -2.25 % for the low and high-altitude bands respectively, which is more than 5 times (low band) and 2 times (high band) larger than the HOMF simulation with homogeneous emissions (see details in Table 4). Since the influence of the emissions is stronger in the lowest levels, the segregation intensity is also larger at lower altitudes. Besides of the mean values, the maximum intensity can reach -50 %, implying that the impact of the heterogenous character of the emissions can be very strong at certain locations, specifically in

the vicinity of the source regions. For the two other negatively correlated reactions, the mean segregation intensities for RH-A and OH are -3.61 % (low) and -3.20 % (high), and those for NO and $RO_2$-A are -9.79 % and -9.21 %. For reaction of RH-B with OH, which has both positive and negative intensities as shown in Figure 7, the mean values are -7.20 % and -10.19 %. Even though the total segregation is characterized by a negative intensity, the maximum positive value of this quantity is higher than 100 %, which occurs near the surface; this may explain why the negative intensity at low altitudes is smaller than the intensity at high altitudes for this reaction. The segregation intensities under the inhomogeneous emission conditions are dependent on the distribution of the emission, so that the numbers cannot be compared directly; however, previous studies also show larger segregation intensities with heterogenous emissions. For instance, Ouwersloot et al. (2011) derive an isoprene segregation intensity of -12.6 % in their heterogeneous case; Kaser et al. (2015) derived from measurements of isoprene local segregation intensities as large as -30 %. Regarding the reaction of NO with $RO_2$-B, the mean segregation intensity at the lower atmospheric levels is negative (-5.15 %) and is positive (0.90 %) at higher levels. The small positive value is probably diminished by the negative intensities; therefore, it is smaller when produced in the HOMF simulation with homogeneous surface emissions.

### 3.3 Influence of the complex terrain

Over the flat terrain, the PBL evolution is mainly controlled by the upward surface heat flux and by the downward heat flux (entrainment) at the top of the PBL. Over the mountains, in addition to the thermally-driven forcing, the advection of the flow, the occurrence of mountain waves, and the rotors also play important roles in the turbulence structure (De Wekker and Kossmann, 2015). In order to assess the role of the turbulence generated by the presence of mountains on the segregation between chemical species, we now consider the topography of the Hong Kong island in our model simulations (simulation TERW, TERE, TERS, TERN). The terrain is expected to affect the chemical reactions by changing (1) the turbulent strength and (2) the mean distribution of the chemical species. We first show these two effects by considering the simulation TERW, in which the prevailing mean winds are westerlies. The horizontal wind velocity and the total TKE (the sum of resolved TKE and SGS TKE) with the PBL height along latitude 22.275 ºN are displayed in Figure 8, which shows how the topography affects the wind field and local turbulence. Compared to the case with flat terrain, the wind speed is smaller at the mountain base and in the valley; while it is significantly larger on the top of the mountains. The derived PBL height mirrors the shape of the terrain (De Wekker and Kossmann, 2015). The PBL height is depressed above the mountain base due to the subsiding circulation associated with the wind system along the slopes; a significant drop in the PBL height is seen over mountain ridges which is attributed to the Bernoulli effect; the PBL is elevated over valleys as the mixed layer is lifted by the horizontal advection from the high windward terrain and the local up-valley wind system. The TKE mainly increases behind the steep hills as a result from the flow separation and shear effect produced by the slope (Cao et al., 2012). The TKE associated with the terrain changes the concentration fluctuations of the chemical species.

The simulated concentrations of chemical species are shown in Figure 9. Comparing to the concentration distributions from the HETF case without topography (Figure 6), the overall patterns are similar when represented on quasi-horizontal terrain-following surfaces. There are, however, more defined structures in the concentration fields when taking into account the influence of the terrain. For the vertical cross sections, since the anthropogenic emissions are located in the urbanized areas around the mountains, the concentrations of NO and ozone are high in the valleys and low over the mountains. The concentrations of biogenic RH-B are highest near the surface of the mountains since this organic species is emitted by the trees located on the slopes of the mountains. The vertical distribution of OH is low near the surface everywhere above the Hong Kong island because it is depleted by both anthropogenic and biogenic species; however, the OH concentration is particularly low in the valleys because the anthropogenic species (CO and RH-A) consume large quantities of OH.

Figure 10 shows the calculated segregation intensities for three different reactions under consideration in this study. As for the mean concentration and the TKE distributions, the structure of the segregation intensities changes with the topography. For the distributions near the ground, it shares patterns similar to those found in the HETF case, since the near-surface chemical concentrations are similar in the two model experiments. For the vertical structures, the topography not only lifts the patterns by the height of the terrain, but it also changes the strength of the segregation intensities. For the reaction between NO and $O_3$, the calculated mean segregation intensity is -3.14 % in the 0-500 m layer and -2.40 % in the 500-800 m layer. These values are similar to those derived in the HETF simulation, implying that the terrain does not substantially change the mean segregation. Because of the complexity of the topography with a large number of mountain ridges and valleys (and hence some possible compensating effects), the resulting influence of the topography appears on the average to be relatively small. However, the local maximum intensity is -59.18 %, which is larger than the -50.44 % obtained in the HETF case. The terrain impact on the reaction of RH-A and OH, and NO and $RO_2$-A is the same to the reaction of NO with $O_3$. For the reaction of OH and RH-B, the averaged intensity for the low altitude band is -5.86 %, and is therefore smaller than the -7.20 % obtained in the HETF case; however, the mean segregation intensity is -11.16 % for the higher altitude band, which is larger than in the HETF calculation. For the reaction between NO with $RO_2$-B, negative segregation intensities are found mainly below 500 m and positive intensities are derived above 500 m. The separation height is raised by about 300 m compared to the HETF case (no topography). The mean intensity below and above 500 m is -5.60 % and 1.65 % respectively.

The concentration fluctuation intensities for the selected species are represented in Figure 11. As defined in Section 2.4, the concentration fluctuation intensity depends on both the mean concentration and the intensity of the turbulence. For the anthropogenic emitted species such as NO with high concentrations in the urban area, the fluctuation generated by the turbulence is relatively large because of the large concentration gradients in this region. The NO concentration fluctuation intensity is highest in the urban regions where fluctuations dominate, for instance, in the valley between the two mountain peaks and at the eastern side of the hills (the leeward slopes). This is consistent with the TKE distribution. In the case of $O_3$, which is consumed by NO, the concentration is low in the urban area; however, the gradient in the concentration is large and

therefore the fluctuation intensity is strong. The situation is similar for OH. For the biogenic species RH-B and its secondary product RO$_2$-B, the calculated concentration fluctuation intensity is low in their source region, implying that the TKE above the mountain is relatively small, and thus the concentration fluctuation intensity is inversely related to the concentration of the two species. As shown in Equation (9), the sign of the segregation is controlled by the correlation factor $r$, and the segregation intensity depends on the concentration fluctuation intensity of both reactants. For the reaction of NO and O$_3$, because the concentration fluctuation intensity for both species is strong in the urban region, the segregation is strong in this area, especially in areas where TKE is large. For the reaction of RH-B and OH, the segregation intensity is more dependent on the concentration fluctuation intensity of RH-B, which has larger values. The segregation intensity for the reaction of NO and RO$_2$-B is controlled by NO at the low altitude, but it is dominated by RO$_2$-B above 500 m. The relationship between the segregation intensity and TKE is shown in Figure 12. For the reaction between NO and O$_3$ and the reaction between NO and RO$_2$-B, the segregation intensities generally increase with TKE, which is similar to the findings of Dlugi et al. (2014). This supports our hypothesis that terrain affects the segregation through the turbulent strength. However, there is no clear relationship between segregation intensity and TKE for the reaction between RH-B and OH. As discussed above, the source of RH-B is located on the top of the mountains where the TKE is relatively small, so the influence of TKE is overwhelmed by the emission distribution.

To better analyse the impact of the topography, the differences of the segregation between the experiment TERW and HETF are shown in Figure 13. The statistics of the differences are listed in Table 5. It shows that, even though the averaged intensities do not change substantially, the local influence of the terrain is important. The maximum differences are 57.95 %, 49.79 %, 154.03 %, 85.73 %, and 186.42 % for the 5 reactions under consideration. For the reaction between NO and O$_3$, the negative intensity increases mainly in the valley near the center and on the east edge of the island, and decreases in the north of the island and on the top of the mountains. The terrain impact on the reaction of NO with RO$_2$-B is similar to what is found for the NO - O$_3$ reaction near the surface, but it is different in the upper layers where the positive segregation becomes stronger. The situation is more complicated for the reaction between OH and RH-B in the surface layer: the strongest influence of the terrain appears at the boundary between the urban and forest emissions where the segregation intensity is largest. For the vertical distribution, the negative segregation decreases at most places in the lower levels, except to the west of the mountains and on the windward side of the slope. At higher altitudes the negative segregation intensity increases.

The differences in the intensities of the concentration fluctuations between the simulations with and without topography are shown in Figure 14. The concentration fluctuation intensities for NO, O$_3$, and OH are all increased in the valleys and on the leeward side of the mountains at the east of the island. This results from the increase in the concentration gradient of these species and from the strong TKE created by the terrain. On the contrary, the concentration fluctuation intensity of RH-B decreases in the region where TKE is large, because the source of the RH-B is not located in an area where turbulence is strong. Thus, the model shows that the concentration fluctuation intensity of a chemical species is dependent on the emissions, the turbulent kinetic energy, and the relative location of the two terms. When we connect the segregation induced by topography

with the differences of the concentration fluctuation intensities, we find that the segregation becomes stronger in areas where the concentration fluctuation intensities of both reactants increase, while it is weaker where their concentration fluctuation intensities decrease. The segregation intensity is dominated by the chemical species that has the largest concentration fluctuation intensity if the changes by the terrain of this latter quantity are opposite for the two reactants.

### 3.4. Impact on the ozone production and destruction

Finally, we estimate the impact of the segregation mechanisms on the formation of a key secondary species: ozone. In our chemical scheme, the limiting chemical processes in the photochemical production of this molecule is provided by the three reactions by which peroxy radicals convert NO into $NO_2$ (see Table 1)

$$R7: HO_2 + NO \rightarrow NO_2 + OH$$
$$R17: RO_2\text{-}A + NO \rightarrow HCHO + HO_2 + NO_2 + REST\text{-}A$$
$$R20: RO_2\text{-}B + NO \rightarrow HCHO + HO_2 + NO_2 + REST\text{-}B$$

Here REST-A and REST-B represent possible secondary products that are not further considered in our simplified analysis. The photolysis of $NO_2$ produced by the three reactions leads directly to the formation of ozone and the production rate of odd
oxygen ($O_x = O_3 + NO_2$), $P(O_x)$, is expressed in a turbulent medium by

$$P(O_x) = k_7(1 + I_7)\langle NO\rangle\langle HO_2\rangle + k_{17}(1 + I_{17})\langle NO\rangle\langle RO_2\text{-}A\rangle + k_{20}(1 + I_{20})\langle NO\rangle\langle RO_2\text{-}B\rangle$$

Here the brackets $\langle\ \rangle$ refer to time averaged concentration at each model grid point. Reaction rates $k_7$, $k_{17}$ and $k_{20}$ and
545 segregation intensities $I_7$, $I_{17}$ and $I_{20}$ correspond to reactions R7, R17 and R20, respectively. Thus, the production rate of odd oxygen is not only determined by the product of the mean concentration of nitric oxide and peroxy radicals but also by the chemical covariance between the fluctuations of these species.

The photochemical loss rate of $O_x$ is due primarily by the following reactions
$$R6: \quad O(^1D) + H_2O \rightarrow OH + OH$$
$$R14: \quad O_3 + OH \rightarrow HO_2 + O_2$$
$$R15: \quad O_3 + HO_2 \rightarrow OH + 2\,O_2$$

with a loss rate in the turbulent medium expressed as

$$L(O_x) = k_6(1 + I_6)\langle O^1D\rangle\langle H_2O\rangle + k_{14}(1 + I_{14})\langle OH\rangle\langle O_3\rangle + k_{15}(1 + I_{15})\langle HO_2\rangle\langle O_3\rangle$$

Figure 15 (upper panels) shows the mean ozone production rate on model level 1 (at the surface following the terrain topography) and at model level 5 (about 200 m above the surface) with the effect of segregation included. A cross section of the same quantity is shown across the Hong Kong island along latitude 22.275 °N. The figure highlights the complexity of the spatial patterns characterizing the formation of odd oxygen with the existence of values generally peaking at the northern and northeastern coasts of the island. A peak in the production is also found in the central valley and on the leeward slopes of the hills. The lower panels of Figure 15 show that the odd oxygen production rate resulting from the segregation effect is reduced by values that can reach more than 60 % locally, specifically along the coasts or in the valleys where the anthropogenic emissions (e.g., traffic) are highest. On the average, the production rate of $O_x$ over the central area ($14 \times 14$ km$^2$) of the inner domain, is reduced by 8 to 12 % in the first 1000 m of the boundary layer (Figure 16). The change in the $O_x$ loss rate is small and positive due to the positive correlation between ozone and OH and between ozone and $HO_2$, but the negative correlation between $O^1D$ and water vapor.

### 3.5 Difference with the wind directions

Additional simulations were performed to assess if the prevailing wind direction affects the segregation intensity since different wind directions are expected to produce different spatial distributions of the turbulence generated by the topography. The calculated statistics are shown in Table 4. For the reaction between NO and $O_3$, the mean segregation intensity in the lower layer is -3.14 %, -3.04 %, -2.78 %, and -2.83 %, respectively for mean winds blowing from the west, east, south, and north directions; the corresponding intensities for the higher altitudes are -2.40 %, -2.45 %, -1.59 %, and -2.84 %, respectively. The largest segregation intensity for the lower level is from the west wind, while that for the higher level is from the north wind. The smallest segregation intensity is from the south wind for both altitude bands. For the reaction of RH-A with OH, the calculated mean intensities for the west, east, south, and north winds are -3.54 %, -3.39 %, -3.15 %, and -3.76 % for the lower altitudes and -3.47 %, -3.41 %, -2.17 %, and -4.39 % for the higher altitudes. The strongest segregation is generated by the north wind and the weakest by the south wind. The same qualitative result is found for the reaction between NO and $RO_2$-A. In this case, the mean intensities are -9.56 %, -9.07 %, -8.63 %, and -10.08 % for the low level, and -9.79 %, -9.51 %, -6.86 %, and -12.32 % for the high level in the four wind directions. For the reaction of RH-B with OH, the mean intensities are -5.86 %, -5.01 %, -6.62 %, and -8.50 % for the four wind directions at the low bands, and -11.16 %, -10.58 %, -10.08 %, and -15.8 % for the high-altitude band. The east wind produces the strongest segregation and the north wind has the lowest intensity at the low level; the south and the north winds give the highest and the lowest segregation intensity at the high level, respectively. For the reaction between NO and $RO_2$-B, the averaged intensities are -5.60 %, -6.03 %, -4.15 %, and -2.91 % at low altitudes, and 1.65 %, 1.04 %, 3.93 %, 5.03 % at high altitudes. The negative intensity at the low level is largest when the wind is from the east, and smallest when the wind is from the north. The positive segregation is strongest for the north wind and weakest for the east wind.

As discussed in the previous section, the large TKE appears at the leeward side of the hills resulting from the flow recirculation and wind shear effect. However, TKE is high only at certain specific locations, while its magnitude is similar for the four experiments in the whole domain. On the other hand, the large TKE is expected to have a strong impact on the chemical reactions at the locations where the two reactants are characterized by large concentration gradients. This means that the

influence of the TKE is dependent on the distribution of the chemical species too. The different wind directions seem to have a small impact on the mean segregation in our simulations, and this is probably because the topography is too complex with a large number of mountain ridges and valleys and therefore the influences are cancelled out when considering the domain as a whole.

## 4 Summary

The large-eddy simulations presented here were performed to study how urban air pollution behaves at the turbulent scale, a scale that cannot be resolved by global or regional models. The segregation effect on the chemical reactions resulting from inefficient turbulent mixing was analysed. The region of the Hong Kong island offers an interesting situation to conduct such a study because of the highly inhomogeneous surface emissions of reactive species and the complex topography.

The inhomogeneity in the emissions tends to increase the segregation intensities by a factor of 2-5 compared to the simulations performed with homogeneous emissions. In some cases, the heterogeneity in the emissions can even generate a change in the sign of the segregation, resulted from the separated sources of the two reactants. In our chemical mechanism, the reaction between NO and $RO_2$-B is not affected by segregation if NO and RH-B emissions are co-located as in the HOMF case, but a separation between the location of the anthropogenic and biogenic emissions produces a segregation between NO and RH-B

at low altitudes, and thus a reduction in the effective rate at which reaction between NO and $RO_2$-B proceeds.

The topography plays a substantial role on the chemical reactions by influencing the turbulence distribution and intensity and hence impacting the concentration distributions. Near the mountains, the TKE increases on the leeward side of the slope, leading to more intense concentration fluctuations for the species with large concentration gradients. For the species with large

spatial concentration gradients in areas where turbulence is weak (e.g. RH-B on the top of the mountain), the intensity of the concentration fluctuation is not enhanced.

The topography has important impact on the segregation intensity locally, especially if the terrain induces strong TKE values in areas where emissions are intense. The differences in the segregation intensities caused by the terrain can be larger than 50

% at certain locations; however, the mean influence of the topography over the whole domain remains relatively small.

Simulations performed for varied wind directions show little differences in the mean segregation intensities, because the values of the TKE generated by the topography are high only at a limited number of locations (generally the leeward side of the mountains) with little impact on the spatially averaged intensity of the segregation.


In the particular setting of our model for the Hong Kong island with heterogeneous emissions and mountainous topography, the averaged segregation intensities from the surface to 500 m is of the order of -3 %, -3.5 %, -6 %, -9.5 % and -5.5 % for the $NO + O_3$, $OH + RH-A$, $OH + RH-B$, $NO + RO_2-A$ and $NO + RO_2-B$ reactions, respectively. At 500 m above the surface, segregation reduces the ozone formation by 8 % compared to a case with complete mixing. At 100 m above the surface, this

reduction is close to 12 %.

The conceptual model presented here and applied to a region with very inhomogeneous conditions highlights the importance of segregation between reactive species, in particular in the convective planetary boundary layer. Segregation, whose intensity is a function of the chemical and turbulent time constants, occurs at spatial and temporal scales that are not resolved by usual

global and regional chemical transport models. The study suggests that, when used in coarse resolution models, the chemical rate constants measured in the laboratory should be adjusted in the boundary layer to correct for the sub-grid segregation effect (see equation 8). This effect is ignored by these models because they assume complete mixing of reactive species within each grid cell.

**Data availability**

The code or data used in this study are available upon request from the corresponding author.

**Author contributions**

YW designed and performed the experiments. YW and YFM analysed the simulations. YW, YFM, and GPB wrote the article. GPB initiated the idea of the work. GPB and TW provided guidance to YW. DME and MB provided advice and support on the setup of the LES. In addition, DME, CWYL, and MB contributed to the editing of the article.

**Competing interests**

The authors declare that they have no conflict of interest.

**Acknowledgements**

This research has been supported by the Hong Kong Research Grants Council (grant no. T24-504/17-N). YFM contribution to this work was supported by the Shenzhen Science & Technology Program (grant no. KQTD20180411143441009). The National Center for Atmospheric Research is sponsored by the US National Science Foundation. We would like to acknowledge high-performance computing support from NCAR Cheyenne.

**Financial support**

The article processing charges for this open-access publication were covered by the Hong Kong Research Grants Council (grant no. T24-504/17-N).

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

**Table 1: The chemical reactions used in the model (The unit of first-order reaction rate coefficients is $s^{-1}$ and that of second-order reaction rate coefficients is $cm^3$ molecule$^{-1}$ $s^{-1}$.**

| No | Reactions | Reaction rates |
|----|-----------|----------------|
| R1 | $NO_2+hv \rightarrow NO+O_3$ | $5.0 \times 10^{-3}$ |
| R2 | $O_3+hv \rightarrow O^{1D}+REST$ | $2.5 \times 10^{-5}$ |
| R3 | $HCHO+hv \rightarrow HO_2+CO+REST$ | $2.5 \times 10^{-5}$ |
| R4 | $HNO_3+hv \rightarrow NO_2+OH$ | $3.0 \times 10^{-7}$ |
| R5 | $O^{1D}+M \rightarrow O_3+REST$ | $0.78084 \times 1.8 \times 10^{-11} \times e^{110/T} + 0.20946 \times 3.2 \times 10^{-11} \times e^{70/T}$; |
| R6 | $O^{1D}+H_2O \rightarrow OH+OH$ | $2.2 \times 10^{-10}$ |
| R7 | $HO_2+NO \rightarrow NO_2+OH$ | $3.7 \times 10^{-12} \times e^{240.0/T}$ |
| R8 | $O_3+NO \rightarrow NO_2+REST$ | $3.0 \times 10^{-12} \times e^{-1500.0/T}$ |
| R9 | $HO_2+HO_2 \rightarrow H_2O_2+REST$ | $2.2 \times 10^{-13} \times e^{600/T} + 1.9 \times 10^{-33} \times C_M \times e^{980/T}$ |
| R10 | $HO_2+HO_2+H_2O \rightarrow H_2O_2+REST$ | $3.08 \times 10^{-34} \times e^{2800/T} + 2.66 \times 10^{-54} \times C_M \times e^{3180/T}$ |
| R11 | $OH+NO_2 \rightarrow HNO_3$ | TROE |
| R12 | $CO+OH \rightarrow HO_2+REST$ | $1.5 \times 10^{-13} \times (1 + 2.439 \times 10^{-20} \times C_M)$ |
| R13 | $HCHO+OH \rightarrow HO_2+CO+REST$ | $5.5 \times 10^{-12} \times e^{125.0/T}$ |
| R14 | $OH+O_3 \rightarrow HO_2+REST$ | $1.7 \times 10^{-12} \times e^{-940.0/T}$ |
| R15 | $HO_2+O_3 \rightarrow OH+REST$ | $1.0 \times 10^{-14} \times e^{-490.0/T}$ |
| R16 | $RH\text{-}A+OH \rightarrow RO_2\text{-}A+REST$ | $1.0 \times 10^{-11} \times e^{-665.0/T}$ |
| R17 | $RO_2\text{-}A+NO \rightarrow HCHO+HO_2+NO_2+REST$ | $2.8 \times 10^{-12} \times e^{300.0/T}$ |
| R18 | $RO_2\text{-}A+HO_2 \rightarrow ROOH\text{-}A+REST$ | $4.1 \times 10^{-13} \times e^{750.0/T}$ |
| R19 | $RH\text{-}B+OH \rightarrow RO_2\text{-}B+REST$ | $1.0 \times 10^{-10}$ |
| R20 | $RO_2\text{-}B+NO \rightarrow HCHO+HO_2+NO_2+REST$ | $1.0 \times 10^{-11}$ |
| R21 | $RO_2\text{-}B+HO_2 \rightarrow ROOH\text{-}B+REST$ | $1.5 \times 10^{-11}$ |

Note: T stands for the temperature; M stands for air; $C_M$ stands for the air density; REST stands for the products that are not evaluated; TROE = $k_1/(1.0+k_2) \times 0.6^{(1.0/(1.0+\log(k_2)2))}$, $k_1 = 2.6 \times 10^{-30} \times (300/T)^{3.2} \times C_M$, $k_2 = k_1/(2.4 \times 10^{-11} \times (300/T)^{1.3})$


**Table 2: List of the numerical experiments.**

| Case name | With terrain | With heterogeneous emissions | Winds (m s$^{-1}$) |
|---|---|---|---|
| HOMF | no | no | West (u=10, v=0) |
| HETF | no | yes | West (u=10, v=0) |
| TERW | yes | yes | West (u=10, v=0) |
| TERE | yes | yes | East (u=-10, v=0) |
| TERS | yes | yes | South (u=0, v=10) |
| TERN | yes | yes | North (u=0, v=-10) |



**Table 3: Dry deposition velocity used in this study.**

| Chemical compounds | No terrain (cm s$^{-1}$) | With terrain (cm s$^{-1}$) | |
|---|---|---|---|
| | | forest | other |
| O$_3$ | 0.06 | 0.6 | 0.01 |
| NO$_2$ | 0.04 | 0.4 | 0.01 |
| HNO$_3$ | 0.5 | 5 | 0.7 |



**Table 4: List of the calculated segregation intensities for the center region (14×14 km$^2$) of the domain.**

| Case name | Altitude (m) | Segregation intensity (%): mean±std (range) | | | | |
|---|---|---|---|---|---|---|
| | | R8: NO+O$_3$ | R16: RH-A+OH | R19: RH-B+OH | R17: NO+RO$_2$-A | R20: NO+RO$_2$-B |
| HOMF | 0-500 | -0.60±0.20(-1.52~-0.12) | -0.84±0.30(-2.46~-0.15) | -5.08±1.82(-12.5~-0.50) | -3.14±1.16(-9.07~-0.53) | 2.77±1.50(-1.17~9.08) |
| | 500-800 | -0.95±0.29(-2.70~-0.25) | -1.52±0.54(-4.97~-0.34) | -8.38±2.68(-22.4~-1.87) | -5.38±1.93(-17.1~-1.27) | 5.43±2.33(0.30~19.0) |
| HETF | 0-500 | -3.21±6.01(-50.44~-0.03) | -3.61±6.13(-47.10~15.52) | -7.20±8.21(-77.13~112.37) | -9.79±13.72(-90.42~-0.43) | -5.15±13.95(-96.73~158.68) |
| | 500-800 | -2.25±3.35(-27.39~-0.16) | -3.20±4.43(-27.49~-0.10) | -10.19±8.60(-50.08~-4.14) | -9.21±9.98(-57.32~-0.69) | 0.90±9.00(-48.81~32.71) |
| TERW | 0-500 | -3.14±6.05 (-59.18~-0.09) | -3.54±6.25 (-53.98~16.75) | -5.86±6.67 (-64.26~188.58) | -9.56±13.78 (-90.40~-0.38) | -5.60±14.67 (-96.41~108.05) |
| | 500-800 | -2.40±3.60 (-41.34~-0.15) | -3.47±4.88 (-37.73~-0.07) | -11.16±9.46 (-52.18~-0.77) | -9.79±10.75 (-72.13~-0.65) | 1.65±10.00 (-67.28~57.11) |
| TERE | 0-500 | -3.04±5.77 (-55.84~-0.06) | -3.39±5.77 (-47.98~6.40) | -5.01±7.35 (-50.73~93.12) | -9.07±13.13 (-89.67~-0.33) | -6.03±13.70 (-95.52~24.90) |
| | 500-800 | -2.45±3.87 (-30.34~-0.14) | -3.41±4.90 (-35.48~-0.10) | -10.58±10.12 (-67.12~-2.75) | -9.51±10.97 (-67.80~-0.78) | 1.04±9.05 (-57.58~41.07) |
| TERS | 0-500 | -2.78±5.36 (-68.79~-0.07) | -3.15±5.47 (-57.74~28.52) | -6.62±8.26 (-73.19~199.54) | -8.63±12.43 (-95.65~-0.10) | -4.15±12.98 (-98.21~121.93) |
| | 500-800 | -1.59±2.62 (-35.05~-0.15) | -2.17±3.18 (-31.34~-0.06) | -10.08±9.63 (-57.11~1.16) | -6.86±7.57 (-65.35~-0.66) | 3.93±7.48 (-50.80~62.30) |
| TERN | 0-500 | -2.83±4.20 (-45.49~-0.08) | -3.76±4.57 (-39.21~8.13) | -8.50±9.21 (-63.65~119.59) | -10.08±10.72 (-88.80~-0.38) | -2.91±12.38 (-94.63~38.86) |
| | 500-800 | -2.84±2.93 (-32.28~-0.11) | -4.39±3.81 (-29.83~-0.09) | -15.80±11.98 (-63.06~-1.38) | -12.32±8.95 (-63.99~-0.51) | 5.03±11.18 (-70.11~55.55) |

**Table 5: List of the calculated segregation differences between the experiments TERW and the HETF run for the center region (14×14 km$^2$) of the domain.**

| Altitude (m) | Segregation intensity (%): mean±std (range) | | | | |
|---|---|---|---|---|---|
| | R8: NO+O$_3$ | R16: RH-A+OH | R19: RH-B+OH | R17: NO+RO$_2$-A | R20: NO+RO$_2$-B |
| 0-500 | 0.10±3.60(-57.95~30.69) | 0.03±3.23(-49.79~23.28) | 0.67±4.97(-98.67~154.03) | 0.12±6.92(-85.73~53.17) | 0.09±8.63(-186.42~59.74) |
| 500-800 | -0.04±1.80(-21.71~15.71) | -0.17±2.03(-22.90~16.58) | -1.06±5.34(-32.40~31.77) | -0.41±4.46(-48.44~31.88) | 1.45±6.57(-60.53~50.21) |

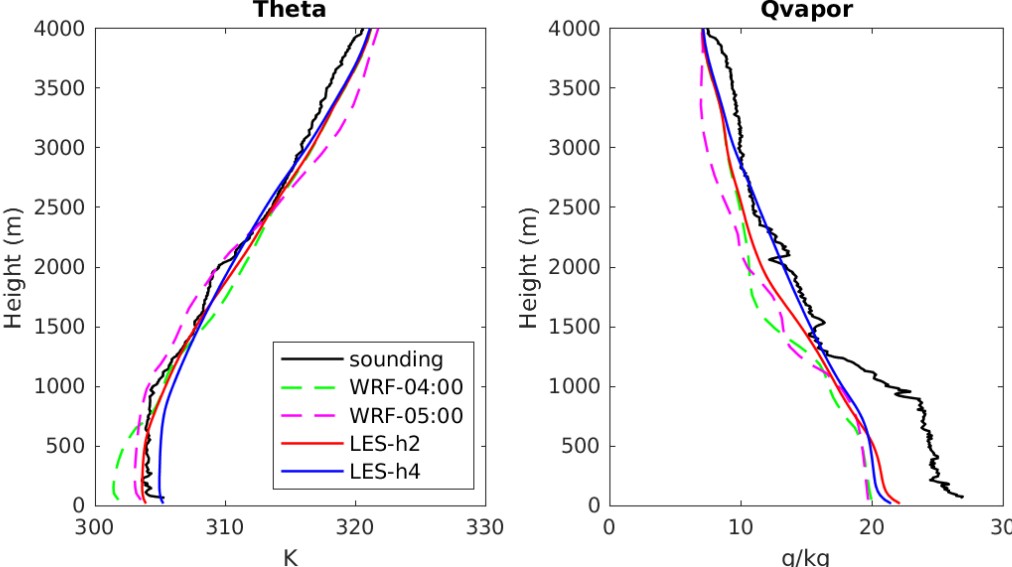

**Figure 1: The evolution of the profiles for potential temperature (left) and water mixing ratio (right) from mesoscale WRF (green dash lines at UTC 04:00; magenta dash line at UTC 05:00) and LES (red solid lines for hour 2 from the start time; bule solid line for hour 4). The measurements at King's Park station taken at UTC 05:00 are shown in black lines (data source: http://www.hko.gov.hk).**

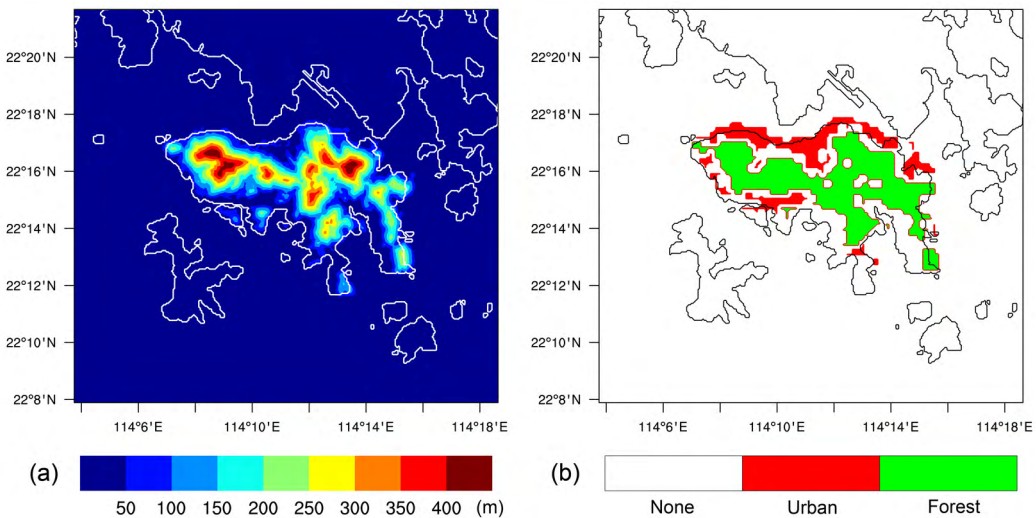

**Figure 2: Elevation of the topography for the Hong Kong island (a) and emission map (b; red is the anthropogenic emission area and green represents the biogenic emission).**

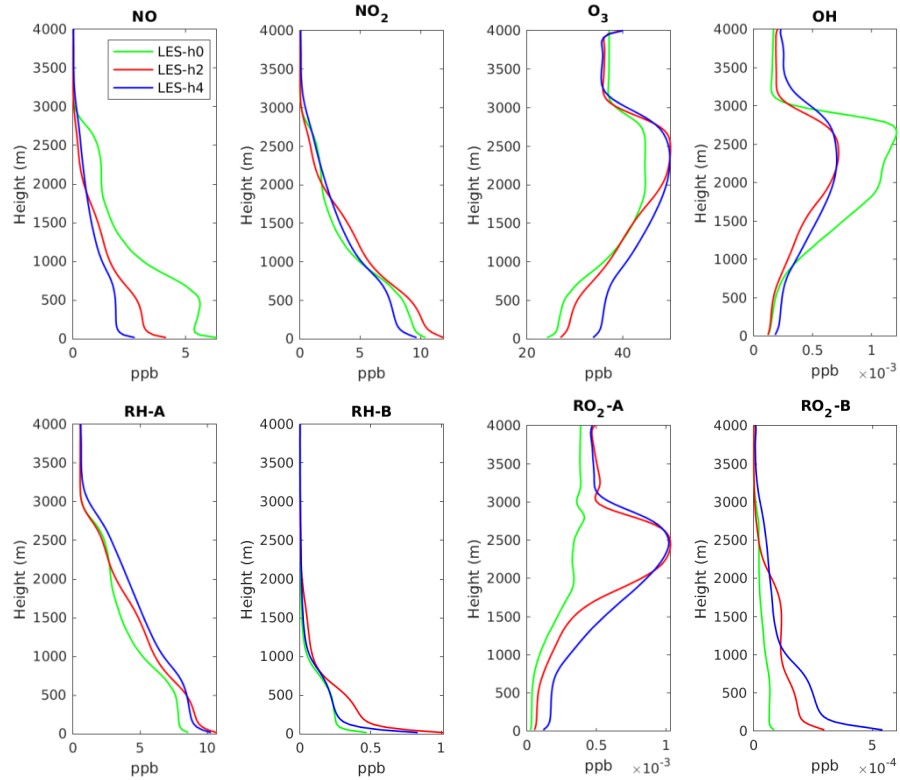

**Figure 3: Domain-averaged profiles of the selected chemical compounds at the initial time (green lines), the hour 2 (red lines), and the hour 4 (blue lines) for HOMF case.**

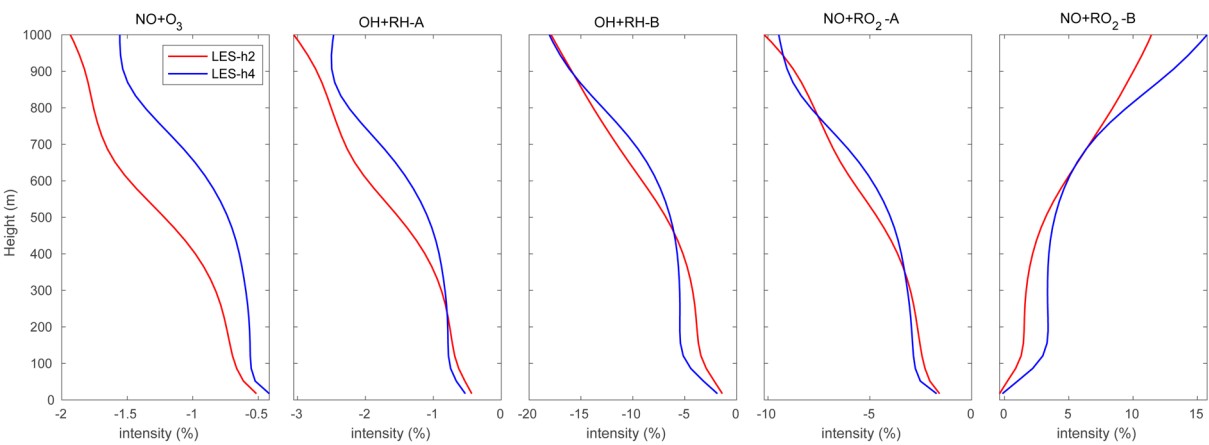

**Figure 4: Domain-averaged segregation intensity profiles of selected reactions at hour 2 (red) and hour 4 (blue) for HOMF case.**

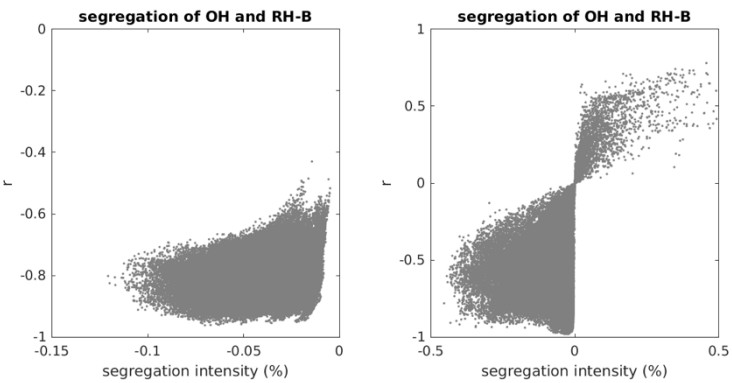

**Figure 5: Relationship between segregation intensity and correlation coefficient (r) for the reaction between OH and RH-B. Left plot is for the simulation HOMF and right plot is for the HETF case.**

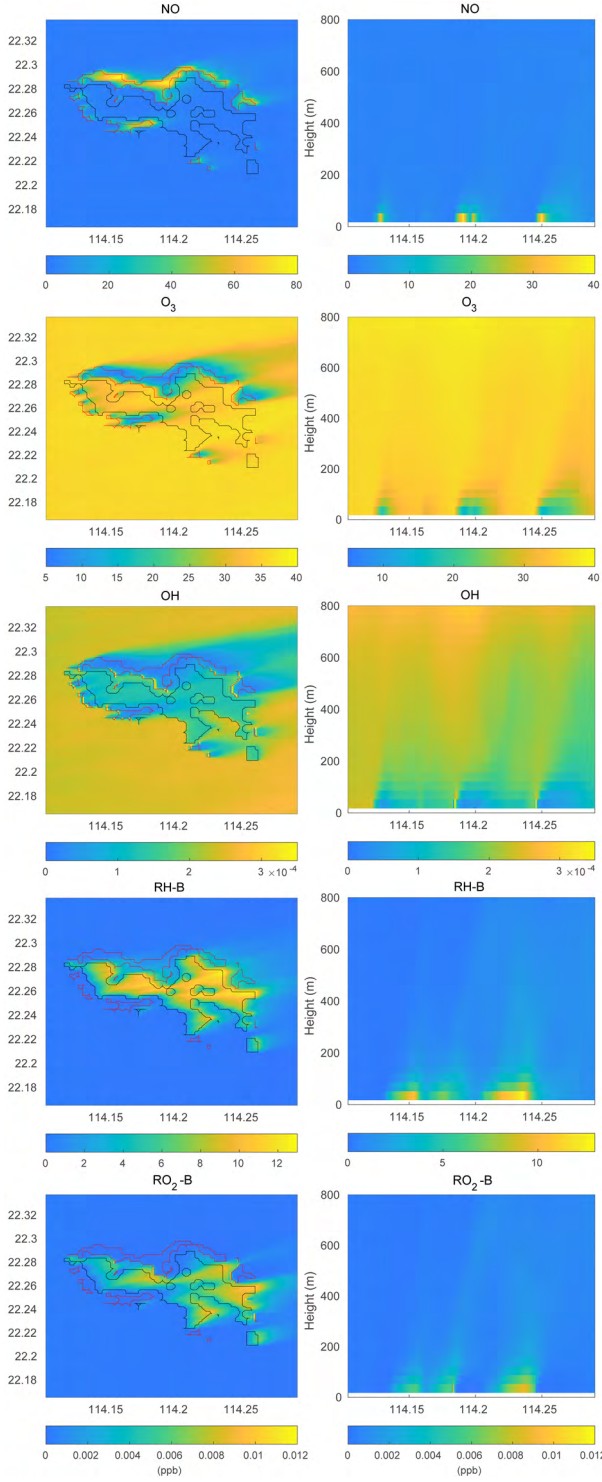

**Figure 6: Volume mixing ratio of the chemical species at the first level (left panel) and vertical cross section along latitude 22.275 °N (right panel) at hour 4 for HETF case. The red line shows the urban area and the black line shows the forest area.**

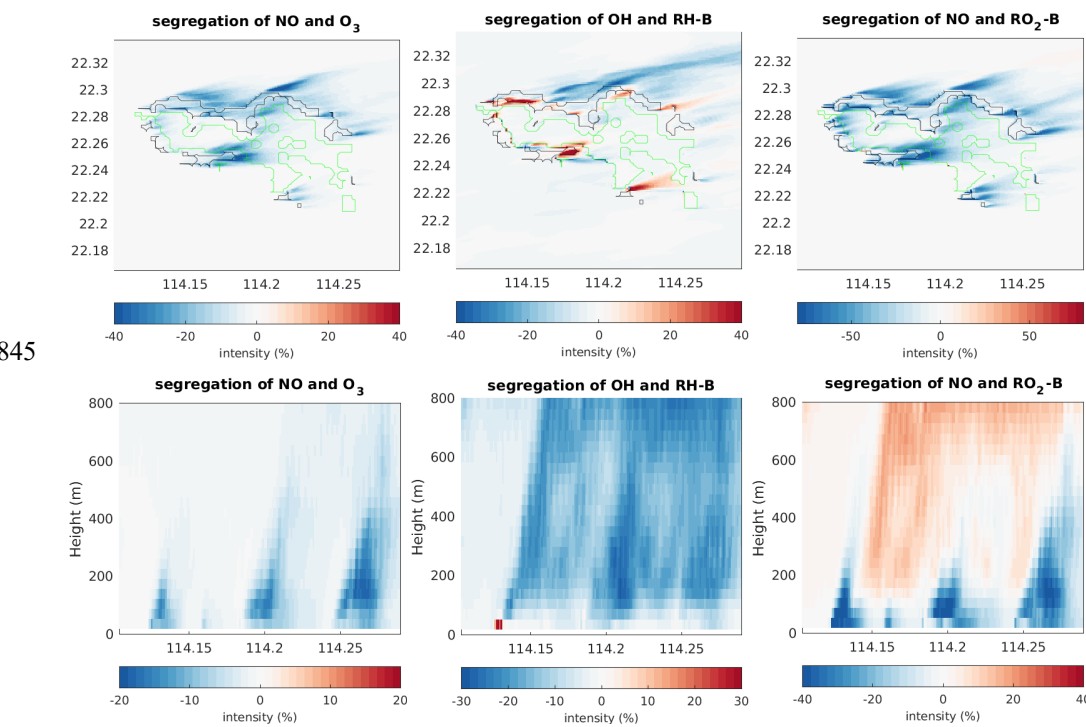

**Figure 7: Segregation intensities at the first level (top panel) and the vertical cross section along latitude 22.275 °N (bottom panel) at hour 4 for HETF case. The black line shows the urban area and the green line shows the forest area.**

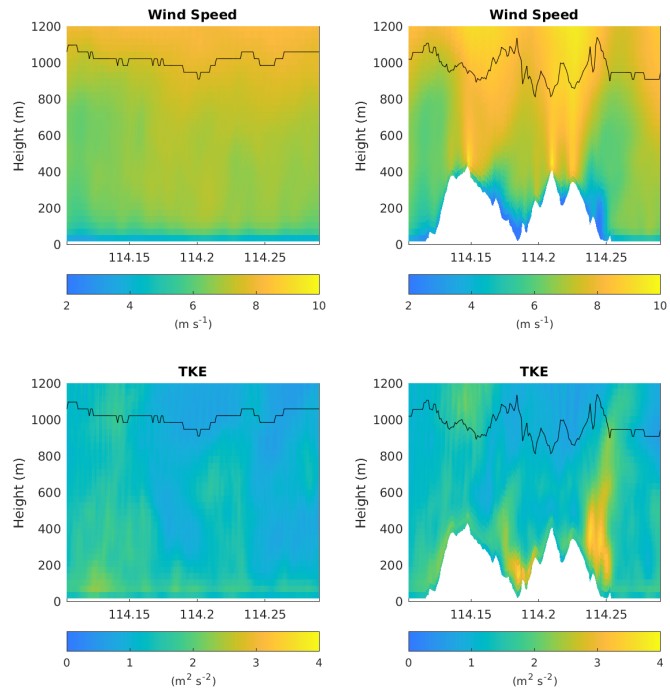

**Figure 8: Vertical cross section of the horizontal wind speed (top panel) and total TKE (bottom panel) along latitude 22.275 °N at hour 4 for simulation without terrain (left column; HOMF/HETF case) and with terrain (right column; TERW case). The black line indicates the PBL height.**

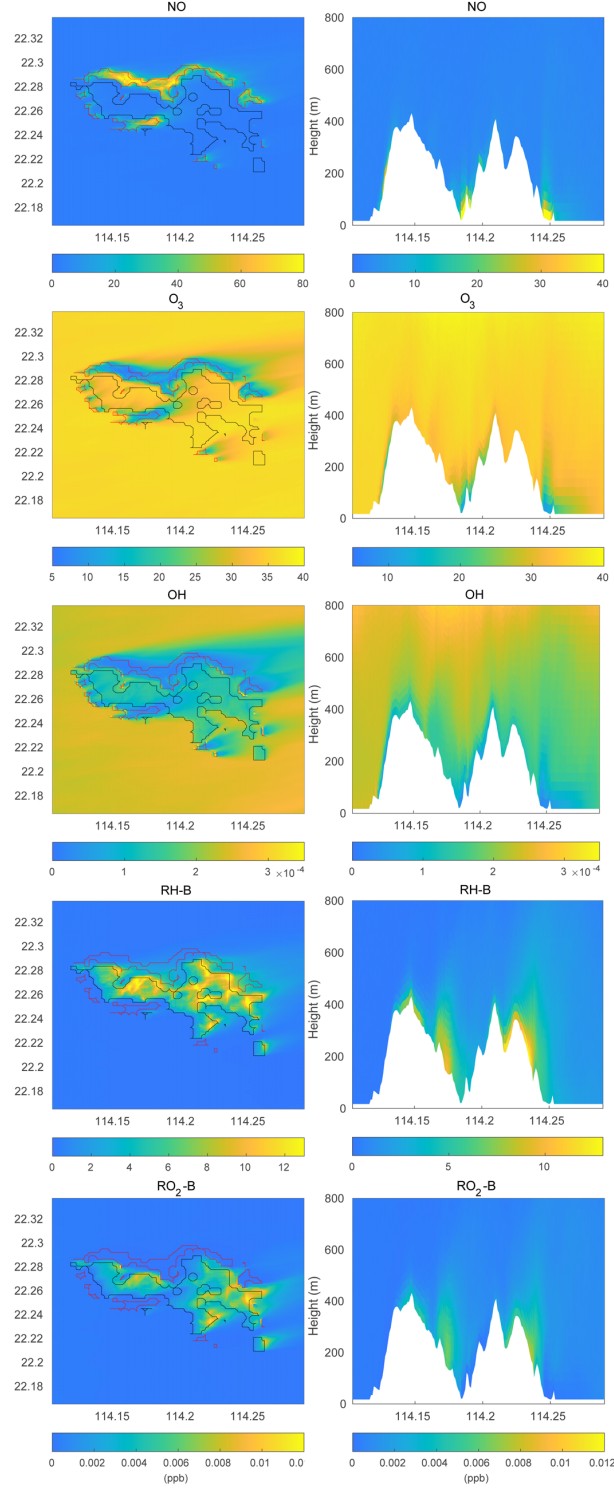

**Figure 9: Same as Fig. 6, but for TERW case.**

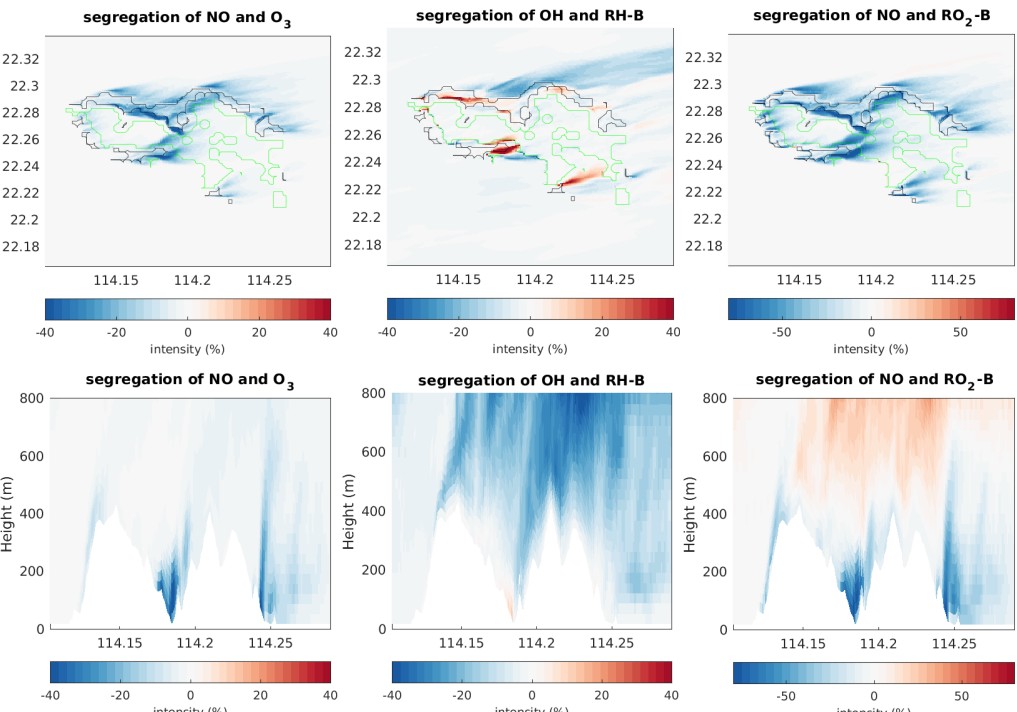

**Figure 10: Same as Fig. 7, but for TERW case.**

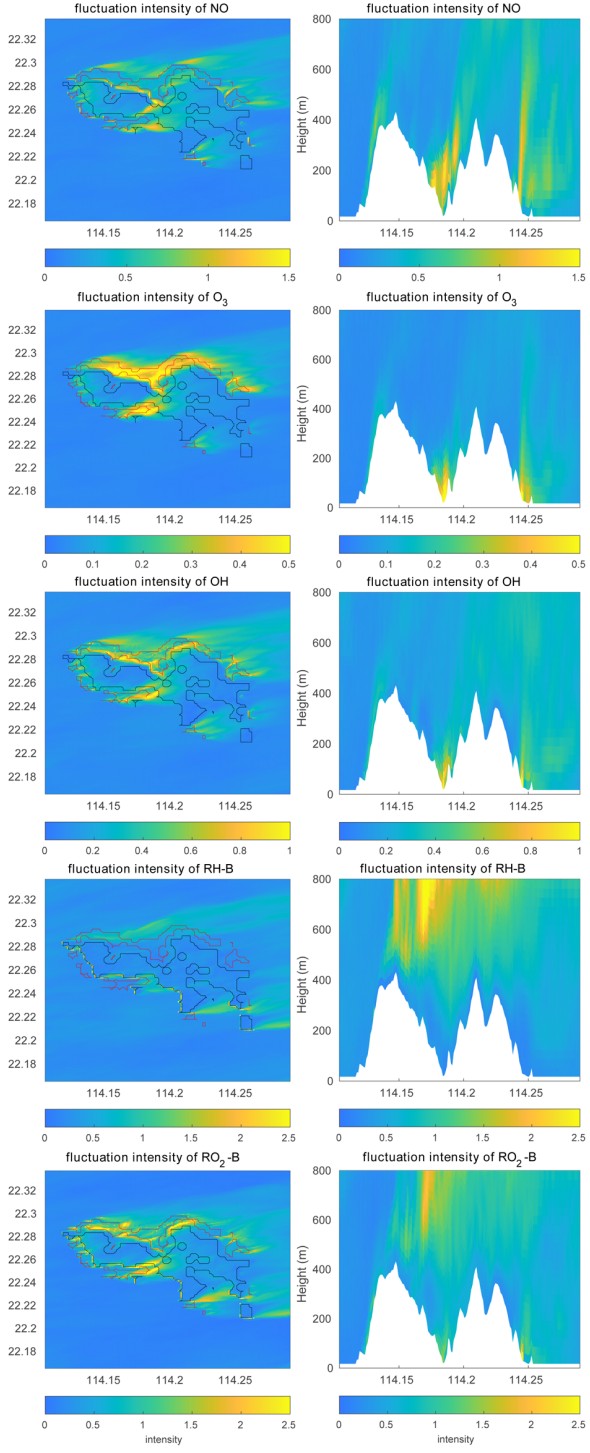

**Figure 11: Concentration fluctuation intensities (variance divided by the mean concentration) at the first level (left panel) and the vertical cross section along latitude 22.275 °N (right panel) at hour 4 for TERW case. The red line shows the urban area and the black line shows the forest area.**

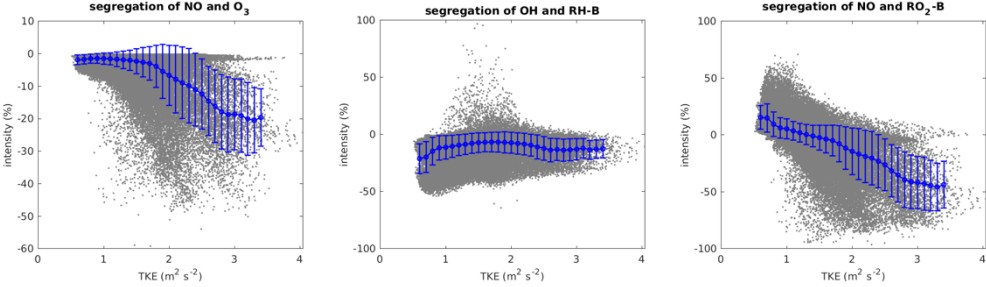

**Figure 12: Relationship between segregation intensity and TKE. The grey dots represent the raw data, and the blue dots are the mean value in every TKE bin with a width of 0.1 m² s⁻². The error bars show the standard deviations.**

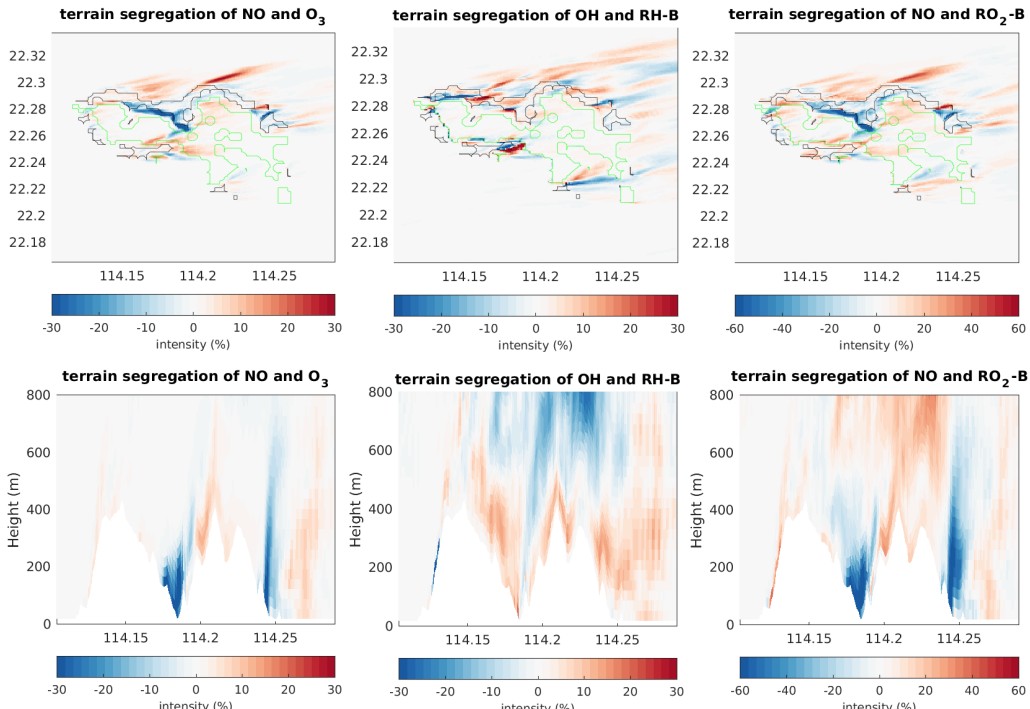

**Figure 13: Differences of the segregation intensity between TERW and HETF (TERW – HETF) at the first level (top panel) and the vertical cross section along latitude 22.275 °N (bottom panel). The black line shows the urban area and the green line shows the forest area.**

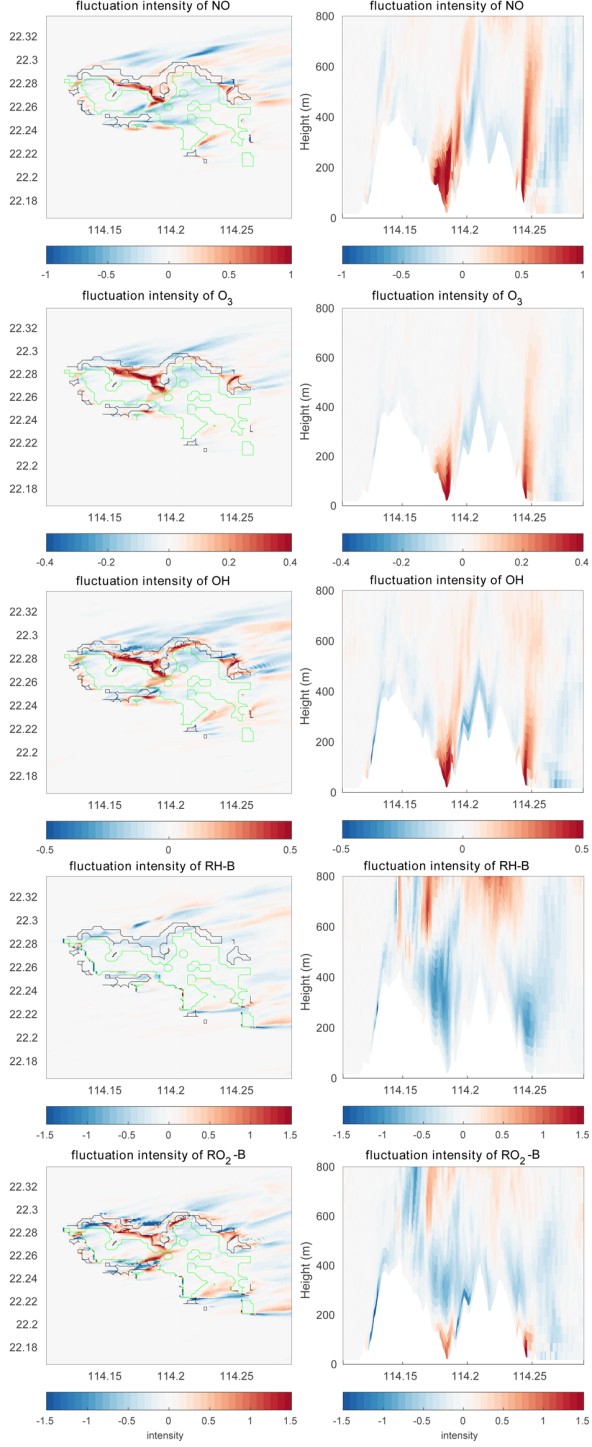

**Figure 14: Differences of the concentration fluctuation intensity (variance divided by the mean concentration) between TERW and HETF (TERW – HETF) at the first level (left panel) and the vertical cross section along latitude 22.275 °N (right panel). The black line shows the urban area, and the green line shows the forest area.**

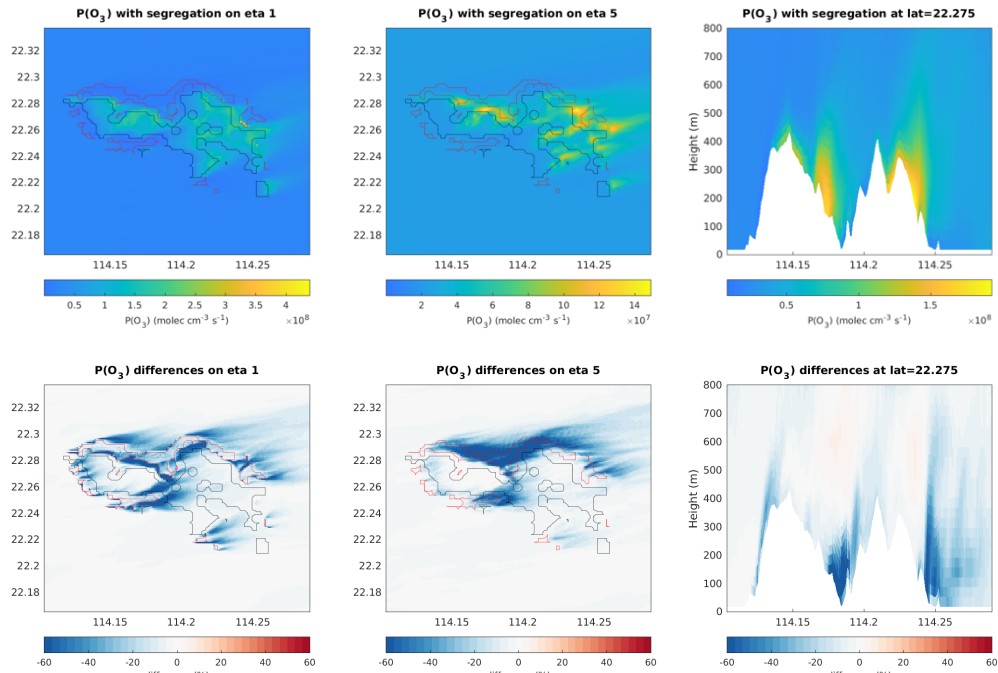

**Figure 15. Upper panels: production rate of ozone (molecules cm⁻³ s⁻¹) in and around the Hong Kong Island. From left to right: values at the lowest model level, values at model level 5 and values as a function of height along the constant latitude of 22.275 °N. Lower panels: Same, but for the percentage difference between ozone production rates with and without segregation effects taken into account.**

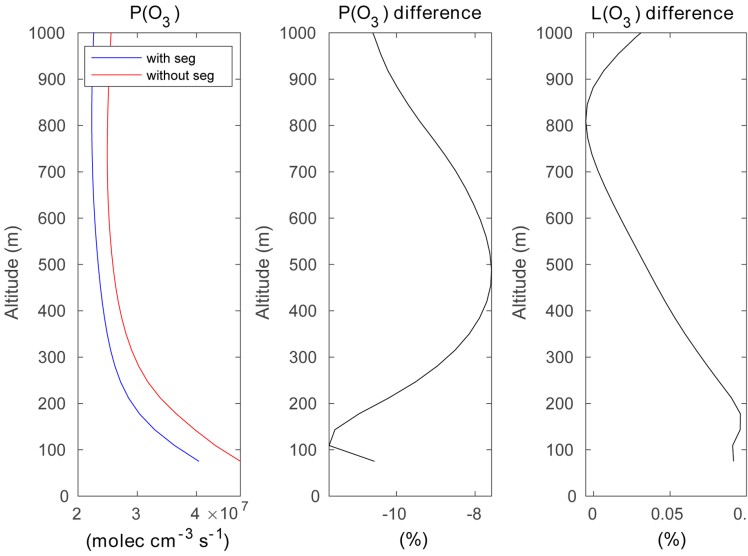

**Figure 16. Left panel: vertical distribution of the ozone production rate (molecules cm⁻³ s⁻¹) averaged over the central 14 x 14 km² of the inner domain with the effect of segregation taken into account (red curve) or ignored (blue curve). Middle and right panels: percentage difference between the two cases for the photochemical production and destruction rates.**