# Peer review of "The impact of inhomogeneous emissions and topography on ozone photochemistry in the vicinity of the Hong Kong island"

_Atmospheric Chemistry and Physics, 2020_

## Referee Comment (RC1) · Anonymous Referee #1 · 27 Oct 2020

This study uses large-eddy simulations to investigate the impact of heterogenous emissions and topography on the segregation of chemical species in the mountain region of the Hong Long island. This is an important topic as global and regional chemical transport models typically cannot resolve subgrid-scale processes and therefore have difficulties accounting for the impact of segregation on chemical reaction rates within the boundary layer. The manuscript is generally well written. Experiments are carefully designed to include different emission, topography, and wind scenarios. Results are also clearly explained. However, a few major flaws need to be clarified before the manuscript can be published on ACP.

[Figure]

- The manuscript only provides a simple review of segregation caused by inhomogeneous emissions. First, please provide a more accurate descriptions of what these studies found. Second, a more thorough review of previous studies, including studies on the impact of terrain, is necessary. Why does this study focus on terrain? What is the role of terrain in regional scales? Third, it is recommended to include a brief description of the main results (e.g., segregation intensity) from previous studies, and elaborate on how the current study differs from previous ones.

- The study uses a flat outer domain, which could cause biases in simulated wind and other meteorological fields. The biases are then passed to the inner domain. If the WRF-LES is used, is it possible to provide more realistic simulations for both the outer and inner domains that apply meteorological fields, typography, emissions, etc from WRF?

- Segregation is important for fast reactions and determined by chemical and turbulent timescales. Therefore, turbulent turn over time and chemical lifetime of the species considered here need to be calculated based on the LES simulations and presented in this study. This study applies a simple chemical mechanism based on Brasseur and Jacob (2017), which implemented a rough categorization of primary anthropogenic and biogenic VOCs. Is this chemical mechanism suitable for the study of segregation? Are these VOCs all reactive? What are the criteria for reactivity in the current study? The lifetimes of VOCs in each category vary largely and are not all less or even comparable to turbulent turn over time. This rough categorization could induce large errors in segregation analysis. Please elaborate more on why this mechanism is selected. Also, please clarify what typical species are included in the anthropogenic and biogenic VOC groups and what the representative lifetimes are for these two groups.

- Because this study investigates segregation in the planetary boundary layer, the manuscript needs to include a paragraph to discuss boundary layer development. Please also elaborate on how terrain influences boundary layer height and whether/how it influences segregation.

- Although the LES is run in an idealized mode in the study, it would be more enlightening to perform some simulation-observation comparisons to assure things are generally consistent with the real world.

- In the result section, the manuscript needs to include a more detailed comparison of the results obtained in this study with those in previous studies. Although the results here are consistent with two studies, how about other studies? Please compare calculated segregation intensities and also justify the differences.

Others:

Lines 16-17: "However, in reality, these species are often segregated due to localized sources and the influence of the topography." It is unclear why topography is referred to here. Is it a finding from previous studies? If so, please add the citations to the introduction section.

Line 40: What are "organized turbulent flows?"

Lines 55-57: Other studies have already investigated the impact of inhomogeneous emissions. How does this study differ from previous ones?

Lines 59-60: What did Kim et al. (2016) find? Low and high NOx conditions could be similar to the mountain and urban regions in this study. Please provide more explanations here. Please fix the typo "NOX."

Lines 60-61: "resulted in?" Please rewrite this sentence with a better clarification. It now reads like it is recommended to remove aqueous-phase chemistry from the LES model... There are multiple obscure sentences in this manuscript. Please double check the writings to avoid misunderstanding.

Lines 65 and 80: Duplicate purposes?

Line 103: Eddies do not produce energy.

Lines 160-164: Please provide citations for the removal rate and the deposition velocities used in this study.

Line 212: Does this sentence make a paragraph?

Lines 232-234: Why is the simulated water vapor from LES higher than in the mesoscale model? Does it generate any or more clouds, which could then influence segregation aloft?

Line 236: Not all the species show similar profiles at hour 2 and 4. So this may not be used to justify chemical equilibrium. Or please elaborate more.

Lines 275-276: Do "hour 2" and "hour 4" represent 14LT and 16 LT? Please use local time instead of hour XX in the main context and the figures. What is "gradual mixing?" Or do the authors mean increased/enhanced mixing?

Line 373: "This is in consistent with ..." Please delete "in."

Lines 373-375: Please rewrite this sentence. . .

Section 3.4: Are wind directions and speeds consistent and constant throughout the whole domain, both horizontally and vertically? Please clarify in this section. Based on the terrain map, north and south winds should be largely different from east and west winds. Please provide more explanations on this.

---

## Referee Comment (RC2) · Anonymous Referee #2 · 4 Nov 2020

1) The paper addresses the topic to study segregation "in order to investigate the degree to which the rates of chemical reactions between two reactive species" are modified compared to the well mixed case. This is done for 5 reactions to study a) the influence of a non-homogeneous distribution of surface sources and b) the influence of topography for a concrete case, the Hong Kong island. They apply an LES embedded in WRF for the regional flow field.

The complete topic is a next step forward if the very systematic study of Ouwersloot et al. (2011) is considered, which is cited. Therefore, the paper represents a substantial contribution to science within the scope of ACP.

[Figure]

2) The application of LES to study segregation is for a concrete landscape in this way is new.

3) The conclusions reached are new with respect to comparable LES studies but will be improved if some additional figures and data (in tables) are added (Remarks).

4) The methods and assumptions seem to be valid but more quantitative information should be given for any reader for better understanding of the results.

5) The interpretations and conclusions need some support by the quantitative presentation of some additional results (Remarks).

6) The authors give credit to related work and indicate their own new contribution. They may also consider: Patton, E.G. et al. (2011): Boundary Layer Meteorol. 100, 91-129. Kramm, G., Meixner, F.X. (2000): Tellus 52A and some literature noted in the remarks.

Remarks:

a) General Remarks: The presentation of segregation intensities $I_S$ is based on the time averaging concept. This is helpful to compare also to results obtained e.g. by:

Verver, G.H.L. et al. (2000), J. Geophys. Res. 105, 3983 - 4002;

Kaser, L. et al. (2015): Geophys. Res. Letter 42, 10.894 - 10.903;

Dlugi, R. et al. (2019): ACPD, as cited; and references in 7)

for the reaction OH + RH-B, which seem to be comparable to OH + isoprene by the rate constant, and for the reacion $O_3$ + NO.

An LES without PBL-scheme is used, with prescribed sensible heat flux $H_S$ , and simplified moisture flux (section 2.2).

The "inner area" has urban area, forest area and "none" (Fig.2). But "none" is sea and sea has $H_S << 220 W m^2$! Please give specific information. The mechanism to produce turbulence should be, therefore, described in detail (A table may help). Please

give also roughness properties for the sea, the urban area and the forest to explain the production of the TKE. Which term of the TKE - balance is dominant? Which processes produce TKE? Please also show wind profiles above sea, urban area, forest (hill). Are there rotating flow elements behind the hill?

What is the influence of bouyancy? You mention "convective conditions" in line 129, but does this really contribute to the TKE in your model setup? Did you specify PBL-height by $\Theta$ - or $\Theta_v$ - profile like in Ouwersloot el. (2011)?

Please formulate a subsection on these details to inform any reader on the physics of the calculations for a better understanding.

b) In line 143-152 you give the emission rates. They are constant for all grid elements of the urban area and the forest. No heterogeneity within the areas is considered. So the scales of heterogeneity are above several hundred meters or more! In line 143 you mention "constant emissions" were used in the outer domain". But here you have sea, a harbour and emission of ships.

Please add maps like Fig.2 for emissions of NO, CO, RH-A and RH-B at the surface for a better understanding by any reader.

c) Line 234: You mention results on water vapour in Fig.1. If water vapour varies it is of direct influence on OH-production (Eq. R6).

d) Line 236: This is not "chemical equilibrium". These are stationary conditions. The ozone production is still visible for 2h and 4h! Note also you have "poluted conditions" while Ouwersloot et al.(2011) has low $NO_x$ (or even no $NO_x$) conditions.

e) You mention several times that high TKE leads to higher (or high) segregation. Please present figures for each reaction to show $I_S$ versus TKE. Also Dlugi et al.(2014), ACP14, 10333-10362 presented their findings for $I_S$ (OH + isoprene) as function of TKE (their Fig.19).

f) For comparison with literature you may also present (for negative and positive $I_S$) $I_S$

as function of the covariance in Eq. 6to test the hypothesis that "$I_S$ is proportional to the covariance" as mentioned by Kaser et al (2015). Dlugi et al.(2019) also presented a figure with r (your Eq. 8) as function of $I_S$ (their Fig.9). Such results may also be added to your presentation to compare with data from literature.

g) Table 1: Dimensions are missing for all quantities.

h) Table 3: Dimensions should be shifted to the right (numbers).

i) Please give the complete notation for "VMR".

j) Fig.6: OH and RH-B: Segregation is given outside the area of emission! (also in Fig 9). How to explain positive $I_S$ ?

h) Please replace the notion "tracer" by "chemical compounds" or "reactants" or another specific word.
* * *

---

## Author Comment (AC1) · 22 Dec 2020

**Response to Interactive comment from Referee #1**

We thank the referee#1 for taking the time to read the manuscript and offer helpful comments and suggestions. We have modified the manuscript according to the referee's comments. The detailed changes can be found in the word-tracking in the manuscript. Additionally, we added one more section (section 3.4 in the new manuscript) to analyse the impact of segregation on the ozone formation, which gives some insight on the coarse models. The point-to-point responses to the referee's comments are listed below. The referee's comment is repeated with our response in bold.

This study uses large-eddy simulations to investigate the impact of heterogenous emissions and topography on the segregation of chemical species in the mountain region of the Hong Long island. This is an important topic as global and regional chemical transport models typically cannot resolve subgrid-scale processes and therefore have difficulties accounting for the impact of segregation on chemical reaction rates within the boundary layer. The manuscript is generally well written. Experiments are carefully designed to include different emission, topography, and wind scenarios. Results are also clearly explained. However, a few major flaws need to be clarified before the manuscript can be published on ACP.

 - The manuscript only provides a simple review of segregation caused by inhomogeneous emissions. First, please provide a more accurate descriptions of what these studies found. Second, a more thorough review of previous studies, including studies on the impact of terrain, is necessary. Why does this study focus on terrain? What is the role of terrain in regional scales? Third, it is recommended to include a brief description of the main results (e.g., segregation intensity) from previous studies, and elaborate on how the current study differs from previous ones.

**Response: We added more description of the segregation studies in the introduction (see detail in word-tracking). Previous studies (e.g. Cao et al., 2012; Rotach et al., 2015; Liang et al., 2020) showed that complex terrain has an important impact on the turbulence in the boundary layer. Since the segregation effect is dependent on the strength of the turbulent mixing, and therefore the segregation intensity is affected by the terrain too. This study applied a concrete landscape in the WRF-LES to study the segregation, which is a step forward of the previous studies. We added more introduction on the role of the terrain in the manuscript.**

- The study uses a flat outer domain, which could cause biases in simulated wind and other meteorological fields. The biases are then passed to the inner domain. If the WRF-LES is used, is it possible to provide more realistic simulations for both the outer and inner domains that apply meteorological fields, typography, emissions, etc from WRF?

**Response: In this study, we focus on the theoretical aspects of the segregation between reactive species in a well-developed PBL. Our goal is not to produce a fully realistic picture of the turbulence in and around Hong Kong. We use the LES methodology to generate turbulence and assess how chemical reactions are affected by a turbulent field. This is an intermediate step towards the development of a more realistic simulation with coupled WRF-Chem and WRF-LES and more realistic emissions over land and over the surrounding**

**ocean. The flat outer domain is used here to generate the meteorological and chemical field for the boundary conditions of the inner domain. We agree that there are biases, but because the periodic boundary condition is used for the outer domain, so there will be biases too if the wind goes over the terrain multiple times.**

- Segregation is important for fast reactions and determined by chemical and turbulent timescales. Therefore, turbulent turn over time and chemical lifetime of the species considered here need to be calculated based on the LES simulations and presented in this study. This study applies a simple chemical mechanism based on Brasseur and Jacob (2017), which implemented a rough categorization of primary anthropogenic and biogenic VOCs. Is this chemical mechanism suitable for the study of segregation? Are these VOCs all reactive? What are the criteria for reactivity in the current study? The lifetimes of VOCs in each category vary largely and are not all less or even comparable to turbulent turn over time. This rough categorization could induce large errors in segregation analysis. Please elaborate more on why this mechanism is selected. Also, please clarify what typical species are included in the anthropogenic and biogenic VOC groups and what the representative lifetimes are for these two groups.

**Response:**
**On purpose, and in order to facilitate our conceptual analysis, we have chosen to use two primary hydrocarbons whose emissions are not co-located: one (RH-A or a surrogate of propane - C3H8) that is emitted in urbanized areas from anthropogenic activities near the coasts, and the second one (RH-B or a surrogate of isoprene ) that is supposed to be from biologic origin and is supposed to be emitted in the forested hills of the Hong Kong Island. The daytime chemical lifetimes between the two different species are very different: 5.5 hours for RH-A (propane-like) and 3 minutes for RH-B (isoprene-like), assuming that the OH concentration is $5 \times 10^6$ cm$^{-3}$. Thus, all VOCs are reactive and the numerical experiment is developed with the purpose of examining the fate of two primary hydrocarbons with a factor 100 different timescales.**
**We added the definition of Damköhler number for the evaluation of the reaction speed in section 2.4. We calculated the turbulent timescale (9min) and the Damköhler number in the result section. The Damköhler numbers are $3 \times 10^{-2}$ in the case of RH-A (relatively slow chemistry) and 0.3 in the case of RH-B (fast chemistry).**

- Because this study investigates segregation in the planetary boundary layer, the manuscript needs to include a paragraph to discuss boundary layer development. Please also elaborate on how terrain influences boundary layer height and whether/how it influences segregation.

**Response: We added more description on the evolution of the PBL in Section 3.1. The plots with PBL height have been added in Figure 8 and more details of the terrain effect on PBL height have been added in Section 3.3. From Figure 8., it is seen that the PBL height is modified by the complex mountainous terrain compared to flat terrain. In flat homogeneous terrain, the PBL is mostly dominated by upward sensible heat flux at the surface and downward sensible heat flux (entrainment) at the top of the PBL. Over the mountainous region, the atmospheric structure becomes much more complicated. In addition to the thermally-driven, the advection of flows also plays an important role for**

the PBL evolution (see e.g., De Wekker & Kossmann 2015). In our case, the type of PBL seems like a contra-terrain following (see Fig.10d in De Wekker & Kossmann 2015). Due to the complex mountainous terrain (many valleys and ridges) and a short time simulation with a simple surface thermally-driven, we didn't find an obvious influence of terrain on the domain-averaged PBL height (the difference less than the vertical resolution). However, our objective in this paper is not to carefully reproduce the development of the boundary layer, but rather to create the turbulent conditions by which inhomogeneous surface emissions would (or not) remain segregated. As a result, we did not consider the influence of PBL height on the segregation. With our focus on the complex terrain and heterogeneous emission sources effects, we performed a series of idealized LES simulations and analyse the results below a fixed height of 800 m, which accounts for more than 80% of the entire PBL.

 - Although the LES is run in an idealized mode in the study, it would be more enlightening to perform some simulation-observation comparisons to assure things are generally consistent with the real world.

Response: In our LES simulations, we are using background concentrations that are consistent with regional patterns of the relatively long-lived species such as ozone or carbon monoxide. From these background initial conditions the LES model calculates the fast variability in the different chemical species including the radicals (OH, HO2, RO2, etc.) for which no observational data are available. Data on species like ozone or carbon monoxide exist at the street level (monitoring stations at selected locations of the city). These are affected by local conditions and it is not appropriate to compare such data with a model that does not represent the details of the local emissions (e.g., evolving traffic) and of the urban canopy.

- In the result section, the manuscript needs to include a more detailed comparison of the results obtained in this study with those in previous studies. Although the results here are consistent with two studies, how about other studies? Please compare calculated segregation intensities and also justify the differences.

Response: We have added a more detailed comparison with previous studies in the Section on model results. In Section 3.1, we added the calculated Damköhler numbers for the selected reactions and compared the Damköhler number of RH-B to previous studies (e.g. Patton et al., 2001; Vinuesa et al., 2005; Dlugi et al., 2019), and they are in good agreement. We also added the relationship between segregation intensity and correlation coefficient as done by Ouwersloot et al. (2011) and Dlugi et al. (2019). In Section 3.2, we added more comparison with Ouwersloot et al, (2011) and Kaser et al. (2015).

Others:
Lines 16-17: "However, in reality, these species are often segregated due to localized sources and the influence of the topography." It is unclear why topography is referred to here. Is it a finding from previous studies? If so, please add the citations to the introduction section.

**Response: We have added more description of the terrain effect from the previous studies in the introduction, and more analysis of the terrain influence on the wind speed, PBLH, and TKE in Section 3.3.**

Line 40: What are "organized turbulent flows?"

**Response: We deleted the "organized".**

Lines 55-57: Other studies have already investigated the impact of inhomogeneous emissions. How does this study differ from previous ones?

**Response: Most earlier studies with chemistry involved focused on forested areas (e.g., investigation of the segregation in relation to the isoprene + OH reaction), and they all used idealized method to generate the heterogeneity of the emissions (Gaussian function by Krol et al. (2000) or simple cut by Ouwersloot et al, (2011)). Our study corresponds to region with heavy-polluted urban conditions surrounding an areas with forested landscape. We also, for the first time, added complex terrain in the segregation study.**

Lines 59-60: What did Kim et al. (2016) find? Low and high NOx conditions could be similar to the mountain and urban regions in this study. Please provide more explanations here. Please fix the typo "NOX."

**Response: Kim et al. (2016) showed that the segregation intensities of isoprene and OH differ at low and high nitrogen oxide ($NO_X$) levels caused by the primary production and loss reactions of OH in different $NO_X$ regime. We added more a detailed description on the previous studies in the introduction. We fixed the "NOX" in the manuscript.**

Lines 60-61: "resulted in?" Please rewrite this sentence with a better clarification. It now reads like it is recommended to remove aqueous-phase chemistry from the LES model. . . There are multiple obscure sentences in this manuscript. Please double check the writings to avoid misunderstanding.

**Response: We have rewritten this sentence. We have checked through the manuscript again to correct the obscure sentences.**

Lines 65 and 80: Duplicate purposes?

**Response: We changed the first "purpose" sentence. This is more like a method paragraph.**

Line 103: Eddies do not produce energy.

**Response: It has been changed to "energy-injection scales".**

Lines 160-164: Please provide citations for the removal rate and the deposition velocities used in this study.

Response: Atmospheric destruction of reservoir species are applied to balance the long-term surface emissions of primary species.  The removal of these reservoirs by photolytic processes in the atmosphere is slow (more than 2 weeks in the case of $HNO_3$ and several days for peroxides). Most of the loss is due to wet removal or dry deposition. We do not simulate an event of strong rain that would remove most of the reservoir species (which are soluble) in just a few minutes. However, in order to keep a balance (stationary state) in the background concentrations and avoid an accumulation of species produced from the ongoing emissions, we applied a loss mechanism with a time scale of about half a day. This may be viewed as the time separating rain events during summer time.
Regarding the dry deposition, we adopted for the gras/forested areas outside the urbanized regions values based on the measurements of Wu et al. (2011) and the analysis of Ganzeveld and Lelieveld (1995). The values were reduced over urban areas. A reference to Wu et al., and Ganzeveld and Lelieveld has been added.

Line 212: Does this sentence make a paragraph?

Response: We have moved this sentence to the end of the last paragraph.

Lines 232-234: Why is the simulated water vapor from LES higher than in the mesoscale model? Does it generate any or more clouds, which could then influence segregation aloft?

Response: The relatively lower water vapour in WRF is most likely related to the decrease of simulated wind speed above the top of PBL. Such weaker wind speed results in a relatively less water vapour transport over the ocean under the steady southwest wind. Therefore, the vertical profiles show a low water vapour abundance in the free atmosphere for lower wind speed (see the figure below). In addition, our LES case is an ideal experiment, which does not distinguish the sea and the land for the underlying surface. We find that there is no any cloud in WRF simulation (the nearest grid of observation site), but LES generates some cumulus with the maximum liquid water mixing ratio (hourly domain average) in the range of 0.018-0.025 g/kg. As reviewer mentioned that the cloud evolution would affect the segregation, but the turbulence process in the cloud is complicated and it's not easy to accurately simulate the clouds both in WRF & LES. This paper mainly focuses on the effects of topography and heterogeneous emission sources on the segregation. Therefore, we are currently not considering the impact of the clouds.

[Figure]

**Vertical profiles of observed and simulated wind speed and total water mixing ratio.**

Line 236: Not all the species show similar profiles at hour 2 and 4. So this may not be used to justify chemical equilibrium. Or please elaborate more.

**Response: We have changed the "chemical equilibrium" to "stationary condition" to avoid misleading.**

Lines 275-276: Do "hour 2" and "hour 4" represent 14LT and 16 LT? Please use local time instead of hour XX in the main context and the figures. What is "gradual mixing?" Or do the authors mean increased/enhanced mixing?

**Response: The "hour 2" and "hour 4" do not represent real time, because we used a fixed surface heat flux to force the development of the PBL, and there is no diurnal variation. The "gradual mixing" here reflects a mixing process changing with time. We have changed it to "enhanced mixing" in the manuscript.**

Line 373: "This is in consistent with ..." Please delete "in."

**Response: Corrected.**

Lines 373-375: Please rewrite this sentence. . .

**Response: We have rewritten this sentence.**

Section 3.4: Are wind directions and speeds consistent and constant throughout the whole domain, both horizontally and vertically? Please clarify in this section. Based on the terrain map, north and south winds should be largely different from east and west winds. Please provide more explanations on this.

**Response: The initial horizontal wind for the different experiments are listed in Table 2. The vertical wind speed is zero initially. In Section 3.4, we only analyse the domain average segregation under different wind directions. The difference are large at certain locations, but after averaging, the differences become smaller. This may be because of the complexity of the topography with a large number of mountain ridges and valleys (and hence some possible compensating effects), so that the resulting influence of the topography appears on the average to be relatively small, and thus the different wind directions have little impact on the mean segregation. We have added this explanation in the manuscript. We plan to do a more detailed analysis in the next study that will be based on more realistic conditions as applied to the Hong Kong island.**

---

## Author Comment (AC2) · 22 Dec 2020

**Response to Interactive comment from Referee #2**

**We thank the referee#2 for taking the time to read the manuscript and offer helpful comments and suggestions. We have modified the manuscript according to the referee's comments. The detailed changes can be found in the word-tracking in the manuscript. Additionally, we added one more section (section 3.4 in the new manuscript) to analyse the impact of segregation on the ozone formation, which gives some insight on the coarse models. The point-to-point responses to the referee's comments are listed below. The referee's comment is repeated with our response in bold.**

1) The paper addresses the topic to study segregation "in order to investigate the degree to which the rates of chemical reactions between two reactive species" are modified compared to the well mixed case. This is done for 5 reactions to study a) the influence of a non-homogeneous distribution of surface sources and b) the influence of topography for a concrete case, the Hong Kong island. They apply an LES embedded in WRF for the regional flow field.
The complete topic is a next step forward if the very systematic study of Ouwersloot et al. (2011) is considered, which is cited. Therefore, the paper represents a substantial contribution to science within the scope of ACP.

2) The application of LES to study segregation is for a concrete landscape in this way is new.

3) The conclusions reached are new with respect to comparable LES studies but will be improved if some additional figures and data (in tables) are added (Remarks).

**Response: We have added more figures (Figure 5., 8., 12., 15.) as suggested.**

4) The methods and assumptions seem to be valid but more quantitative information should be given for any reader for better understanding of the results.

**Response: We have added more information according to the reviews' comments (see the responses to the remarks below and the word-tracking in the manuscript).**

5) The interpretations and conclusions need some support by the quantitative presentation of some additional results (Remarks).

**Response: More detailed interpretations of the results were added in the text (see the responses to the remarks below and the word-tracking in the manuscript).**

6) The authors give credit to related work and indicate their own new contribution. They may also consider: Patton, E.G. et al. (2011): Boundary Layer Meteorol. 100, 91-129. Kramm, G., Meixner, F.X. (2000): Tellus 52A and some literature noted in the remarks.

**Response: We have added more details of the methods, assumptions and results analysis to help the readers for better understanding. We have referred to the conceptual studies of Patton et al. (2001) and Kramm and Meixner (2000). For the changes in the manuscript, see the word-tracking.**

Remarks:

a) General Remarks: The presentation of segregation intensities IS is based on the time averaging concept. This is helpful to compare also to results obtained e.g. by:

Verver, G.H.L. et al. (2000), J. Geophys. Res. 105, 3983 - 4002;

Kaser, L. et al. (2015): Geophys. Res. Letter 42, 10.894 - 10.903;

Dlugi, R. et al. (2019): ACPD, as cited; and references in 7)

for the reaction OH + RH-B, which seem to be comparable to OH + isoprene by the rate constant, and for the reacion O3 + NO.

**Response: We have added more comparisons with these studies in the section on results (Section 3.1 for homogeneous case and Section 3.2 for inhomogeneous case) with references to the suggested authors.**

An LES without PBL-scheme is used, with prescribed sensible heat flux HS , and simplified moisture flux (section 2.2).

The "inner area" has urban area, forest area and "none" (Fig.2). But "none" is sea and sea has HS << 220Wm2! Please give specific information. The mechanism to produce turbulence should be, therefore, described in detail (A table may help). Please give also roughness properties for the sea, the urban area and the forest to explain the production of the TKE. Which term of the TKE - balance is dominant? Which processes produce TKE? Please also show wind profiles above sea, urban area, forest (hill). Are there rotating flow elements behind the hill?

What is the influence of bouyancy? You mention "convective conditions" in line 129, but does this really contribute to the TKE in your model setup? Did you specify PBL-height by Θ - or Θv - profile like in Ouwersloot et el. (2011)?

Please formulate a subsection on these details to inform any reader on the physics of the calculations for a better understanding.

**Response: In order to focus on the terrain effect, the land-sea differences are not considered in this study. The surface of the entire domain is set as the land (there is no sea) with a fixed sensible heat flux of 230 W/m². This explanation has been added in the manuscript to avoid misunderstanding. The roughness of the different underlying surfaces for urban area and the forest is indeed important for the momentum flux and shear production of TKE, but in our case only a small area around the Island is urban region and to better distinct the effects of heterogeneous emission sources, a constant roughness length for land is used. In our case, the TKE is mainly produced by the buoyancy and shear. We didn't analyse the TKE balance or which term is dominant for our case because it's not the focus of this study. But we added the relationship between the TKE and segregation intensity in the section 3.3, see figure 11 in the manuscript. We have added the wind speed cross-section plots in Figure 8 to show the wind at different locations.**

**In our LES case, the buoyancy flux ($w'\theta_v'$) is from the surface sensible heat flux and moisture fluxes (similar as Ouwersloot et al., 2011). We set a fixed sensible heat flux of 230 W/m2 at the surface; while the latent heat flux was calculated in the model from a prescribed surface water vapor (see section 2.2). In our case, the buoyancy flux is also dominated by the kinematic sensible heat flux $w'\theta'$, which is almost 4 times of the kinematic moisture heat**

flux (see figure below; the kinematic sensible heat buoyancy flux is 0.203, the kinematic latent heat buoyancy flux in the range of 0.048~0.054).

[Figure]

Domain-averaged surface buoyancy flux: $\langle w'\theta_v' \rangle \approx (1 + 0.61q)\langle w'\theta' \rangle + 0.61\theta\langle w'q' \rangle$
(see the formula from Ouwersloot et al., 2011)

Our case is a convective boundary layer, the vertical profiles of the virtual potential temperature $\theta_v$ do not present clear gradients, so we can't detect the PBL height well with the common method of $\theta_v$ gradient. This feature also can be seen in Ouwersloot et al. (2011). We tried several methods, e.g., maximum gradient of $\theta_v$ or total water mixing ratio, bulk Richardson number, the location of 90% of the maximum wind speed, and the maximum coverture of $\theta_v$. The PBL heights from different methods are inconsistent, but they all can reflect the influence of terrain on the PBL height. By comparing with the results of mesoscale model WRF (PBLH variable in wrfout files), we found that modified $\theta_v$ gradient method (the first level with $\frac{\partial \theta_v}{\partial z} = \frac{\theta_v(z)-\theta_{v_s}}{z-z_s} > 0$, where $\theta_{v_s}$ is surface virtual potential temperature & $z_s$ is the surface altitude) and bulk Richardson number method (the first level with $R_{ib} = \frac{g}{\theta_{v_s}} \frac{[\theta_v(z)-\theta_{v_s}][z-z_s]}{[u(z)-u_s]^2+[v(z)-v_s]^2} = 0.25$, the subscript "s" represents surface) are consistent with WRF output PBLH. We have chosen the modified $\theta_v$ gradient method for the PBLH calculation and added the description in Section 3.1. We have also added the PBL height in Figure 8 to show the influence of the terrain. Due to the complex influences of the terrain on the PBL height, we mainly focus on analysing the results below 800 m (which accounts for about 80% of the entire PBL).

b) In line 143-152 you give the emission rates. They are constant for all grid elements of the urban area and the forest. No heterogeneity within the areas is considered. So the scales of heterogeneity are above several hundred meters or more! In line 143 you mention "constant emissions" were used in the outer domain". But here you have sea, a harbour and emission of ships.
Please add maps like Fig.2 for emissions of NO, CO, RH-A and RH-B at the surface for a better understanding by any reader.

Response: The purpose is not to perform a fully realistic simulation taking into account the detailed emissions from harbours and individual ships, but to simulate a simple case at medium resolution (100 m) to assess how anthropogenic and biogenic emissions localized in different areas of the island contribute to the formation of ozone and other secondary species. Thus, we designed numerical experiments as follows: the inner domain (Hong Kong Island) that includes topography and inhomogeneous emissions is surrounded

**by a flat outer domain area with homogeneous emissions that serves as a broad "boundary condition". In the inner domain, the emission rates for the anthropogenic species (NO, CO and RH-A) are uniform all over the "urban region" (mostly the edges of the island) and zero over other areas; and the emissions of biogenic RH-B are uniform over the forested hills (inside areas of the island) and zero elsewhere. This is shown in Figure 2. Separate emission maps for NO, CO, RH-A, and RH-B will not give more information in our case.**

c) Line 234: You mention results on water vapour in Fig.1. If water vapour varies it is of direct influence on OH-production (Eq. R6).

**Response: The reaction R6 uses the water vapor content from the model to calculate the OH concentration, so the water vapor should have direct influence on OH production, but the detail of the influence is not considered in this study.**

d) Line 236: This is not "chemical equilibrium". These are stationary conditions. The ozone production is still visible for 2h and 4h! Note also you have "polluted conditions" while Ouwersloot et al.(2011) has low NOx (or even no NOx) conditions.

**Response: We changed "chemical equilibrium" to "stationary condition" in the context. We agree that we made our calculations under polluted conditions for urban area, which is also a main difference from previous studies focusing on isoprene chemistry.**

e) You mention several times that high TKE leads to higher (or high) segregation. Please present figures for each reaction to show IS versus TKE. Also Dlugi et al.(2014), ACP14, 10333-10362 presented their findings for IS (OH + isoprene) as function of TKE (their Fig.19).

**Response: We have added the plots of TKE versus segregation for the simulation TERW in Figure 12. From this graph, we see that the segregation intensities increase with TKE for most reactions except for RH-B + OH. The lack of clear relationship between TKE and the segregation of RH-B + OH, might be due to the fact that RB-B is emitted on the top of the mountain where the TKE is small. Dlugi et al. (2014) showed that the segregation of isoprene + OH increases with TKE for campaigns in the forest. Our results are more complicated in exist of the terrain.**

f) For comparison with literature you may also present (for negative and positive IS) IS as function of the covariance in Eq. 6 to test the hypothesis that "IS is proportional to the covariance" as mentioned by Kaser et al (2015). Dlugi et al.(2019) also presented a figure with r (your Eq. 8) as function of IS (their Fig.9). Such results may also be added to your presentation to compare with data from literature.

**Response: We plotted the relationship between segregation intensity and r in Figure 5, and have added some comparison with the previous studies in the manuscript.**

g) Table 1: Dimensions are missing for all quantities.

**Response: We have added the unit in the table.**

h) Table 3: Dimensions should be shifted to the right (numbers).

**Response: It has been changed.**

i) Please give the complete notation for "VMR".

**Response: Corrected.**

j) Fig.6: OH and RH-B: Segregation is given outside the area of emission! (also in Fig 9). How to explain positive IS ?

**Response: From the vertical cross-section, we can see that the segregation intensity of RH-B and OH is small at the surface layer over the forest, so the segregation seems to be outside the area of emission. The positive segregation happens when the RH-B and OH are correlated. From Fig 5. RH-B and OH are both high at the edge between forest and urban area, leading to the positive correlation and thus positive segregation.**

h) Please replace the notion "tracer" by "chemical compounds" or "reactants" or an- other specific word.

**Response: Corrected.**

---

## Author Response (AR2)

**Response to reviewers' comments**

**We thank the editor and reviewers for taking time to read the revised manuscript and our responses to the comments for ACPD. Our answers to the new comments are listed below. Each reviewers' comment is repeated with our response in bold.**

Reviewer 1:

1. In the revised manuscript, it states "With an OH concentration of 5 x 10^6 molecule cm-3, the corresponding chemical lifetimes of RH-A and RH-B are approximately 2 days and 30 minutes, respectively." However, in the response to reviewers, it states "The daytime chemical lifetimes between the two different species are very different: 5.5 hours for RH-A (propane-like) and 3 minutes for RH-B (isoprene-like), assuming that the OH concentration is 5 x 10^6 cm-3." Please clarify the differences and how they are calculated.

**Response: We are sorry for the mistake in the response. The values in the manuscript are the correct ones. The chemical lifetimes of RH-A and RH-B are calculated by their reaction rate constants with OH (k) and the OH concentration ([OH]): $1/(k \times [OH])$. We have added this equation in the manuscript.**

2. The manuscript now provides a good review of previous studies in the introduction section. It would be helpful to include more of these studies in the comparison in the result section. Studies such as Krol et al. (2000), Li et al. (2016), and Kim et al. (2016) also calculated Damköhler number and/or intensity of segregation. Also, besides simply compare the values, please provide brief explanations about why they are consistent or different.

**Response: We have added more comparison to the previous studies in the result section as the reviewer suggested. Some explanations are added too. Please see the track-change in the manuscript.**

3. It is necessary to acknowledge the difficulties in representing PBL processes and cloud convection in WRF. Even though the modified $\theta$v gradient method or other methods used in the study were consistent with PBL from WRF, it doesn't guarantee the results are realistic, especially without comparing with observations. It would be helpful to add the following citations which studied the gaps in different ways.
Barth et al., Atmos. Chem. Phys., 7, 4709–4731, 2007
Li et al., Atmospheric Environment 199, 88–101, 2019
Mapes et al., J. Atmos. Sci., 61(11), 1284–1295, 2014

**Response: We have addressed the difficulties in representing PBL processes and cloud convection in mesoscale WRF as suggested. The recommended references are added too. See the track-change in the manuscript.**

4. For Table 1, please explain why some reaction rates have been revised. Have the authors redone all the simulations using these updated rates?

**Response: We have not revised any reaction rates in our revised study, but we discovered errors in the display of the value of these rate constants in Table 1. In the original manuscript, we had typed the formula for some reaction rates in a wrong line of the table. We made therefore the appropriate in the revised manuscript and restored the values as used in our model..**

Reviewer 2:
Only one typing error remains in
line: 231: It should read "2.5 * 10 ^{-5} s^{-1}".

**Response: We have corrected it.**

[revised manuscript text omitted]